# LIDAR-assisted nonlinear output regulation of wind turbines for fatigue load reduction

Robert H. Moldenhauer[1] and Robert Schmid[1]

[1]Department of Electrical and Electronic Engineering, University of Melbourne

**Correspondence:** Robert H. Moldenhauer (moldenhauer.r@student.unimelb.edu.au)

**Abstract.** Optimizing wind turbine performance involves maximizing or regulating power generation while minimizing fatigue load on the tower structure, blades, and rotor. In this article, we explore the application of a novel turbine control methodology referred to as nonlinear output regulation (NOR) for improving turbine control performance. NOR constructs a torque balance equation under which the closed loops follows desired stable dynamics, and solves it for the generator torque in partial load operation and for the blade pitch angles in full load operation, in a unified manner across both operating regions. The regulation relies on an estimate of rotor-effective wind speed. We consider estimation based on the turbine's SCADA, in particular the Immersion & Invariance (I&I) estimator, as well as LIDAR. Furthermore, we propose to use an average of the I&I and LIDAR estimates, where the LIDAR buffer time is chosen to compensate I&I's delay, to obtain a real-time low-variation estimate.

The performance of the NOR controller with the averaged I&I and LIDAR estimate is compared against a state-of-the-art baseline reference controller known as ROSCO in both its standard feedback-only configuration as well as an existing LIDAR-assisted control version of ROSCO that uses a pitch feedforward. NOR with the averaged I&I and LIDAR estimate matches this LIDAR-assisted ROSCO's rotor speed tracking improvements over feedback-only ROSCO, but also significantly reduces fatigue loads and actuator usage. In particular, the blade flapwise damage-equivalent loads reduction corresponds to a doubled lifespan, and pitch rate is reduced by more than a third. The reductions are achieved without sacrificing power generation.

## 1  Introduction

Wind energy conversion systems are one of the most cost-effective and widely used sources of renewable energy. A 2021 study by Fraunhofer ISE (Bundesverband der Energie- und Wasserwirtschaft, 2023) put the levelised cost of energy (LCOE) for wind energy in Germany at 3.94-8.29 eurocents per kWh, which is about half the LCOE of lignite, hard coal and gas turbines (Kost et al., 2021). In 2022 wind energy accounted for 21.7% of energy production in Germany. The fastest growing market is China, where market volume of wind energy is expected to grow from $443.74\,\text{GW}$ in 2023 to $772.64\,\text{GW}$ in 2028 (Mordor Intelligence, 2024).

Over recent years, upwind, three-bladed, horizontal-axis variable-speed and variable-pitch wind turbines have become the predominant type (Gambier, 2022). The grid connection of variable-speed turbines is realized with power converters. With electronics the generator torque can be controlled, which affects the energy that is extracted from the rotor, and therefore how much it is accelerated or decelerated. Variable-pitch refers to the availability of blade pitch actuation, with the main purpose of

decreasing energy capture at high wind speeds for safety. The primary control objective is to maximize power capture at lower wind speeds, and to not excessively exceed rated power at higher wind speeds. This can be achieved with fairly simple control methods, and more sophisticated methods can provide only marginal power capture improvements. An important secondary control objective is fatigue load reduction. As wind is inherently turbulent, tower and blades are subject to ever-changing stress and strain, which causes mechanical fatigue. Furthermore, the blade pitch actuation is susceptible to wear and tear. These factors are the reason why wind turbines generally have a lifespan of only around 20 years. Therefore, the reduction of fatigue loads and actuator usage through control has the potential to make wind turbines more durable and further decrease their LCOE.

Control methods for improving turbine performance have been subject of extensive research for at least five decades, and summaries are available in the monographs (Bianchi et al., 2007; Munteanu et al., 2008; Luo et al., 2014; van Kuik and Peinke, 2016; Gambier, 2022), for example. For a very recent survey on control methods for wind turbine fatigue load reduction, see (Yaakoubi et al., 2023). Such methods may be broadly separated into strategies for life-of-turbine operating methods that seek to improve turbine life-span by derating, or even shutting down, the turbine in certain operating conditions, and real-time control methods that seek to achieve maximum power point tracking (MPPT) or rated power generation while designing the control actuation in a manner that reduces strain on the tower, blades and generator. Papers in the first category include (Bech et al., 2018), who sought to identify extreme precipitation events causing the largest impact on turbine blade fatigue loads. Operational strategies to find suitable trade-offs between fatigue loads and power production in the long-term management of a wind farm have been investigated in (Requate et al., 2023) and (Kipchirchir et al., 2023).

In this paper we give our attention to the second category of controller methodologies, in which conventional control methods for MPPT are augmented to reduce fatigue loads. According to a 2016 review about the state of the art and future challenges of wind turbine control, the industry standard is to use single-input-single output (SISO) gain-scheduled PID-regulators (van Kuik and Peinke, 2016, Chapter 4). The Reference Open-Source Controller (ROSCO) (Abbas et al., 2022) was recently developed to provide a modular reference wind turbine controller that represented industry standards and was shown to provide better performance than existing reference controllers, in particular the baseline controller of Jonkman et al. (2009).

Many researchers have investigated control methodologies that use an estimate of the wind speed. Wind speed estimation can be classified into two approaches. The first approach uses the rotor speed measurement and other data available from the turbine's SCADA to compute the rotor effective wind speed (REWS), a single wind speed value representing the equivalent steady, horizontal and uniform upstream velocity. Many strategies, such as the extended Kalman filter, Immersion & Invariance (I&I) estimator, and power balance estimator, have been investigated to compute the REWS estimate. See (Soltani et al., 2013) for a comparison of estimation methods.

The second approach involves external wind measurement devices, most commonly involving light detection and ranging (LIDAR) sensors, which uses lasers to measure wind speeds in front of the turbine. A detailed description of LIDAR for wind turbines can be found in (Schlipf, 2016). LIDAR-assisted control has been shown to successfully reduce fatigue loads in (Schlipf et al., 2013; Fu et al., 2023).

One popular control methodology enabled by wind speed estimation is disturbance accommodation control (DAC), also known as disturbance tracking control (DTC), which was introduced by (Balas et al., 1998). DAC uses a superposition of a stabilizing feedback term and a feedforward of the disturbance estimate, aimed at disturbance rejection. If control and disturbance are matched, an equation can be solved to find a feedforward gain, which immediately compensates the disturbance. In wind turbines, drivetrain and blade pitch actuation dynamics, albeit being relatively fast, make this equation unsolvable directly. It is therefore solved approximately (Yuan and Tang, 2017), (Yaakoubi et al., 2023). DAC was also featured in (Wright, 2004) and (Wright and Fingersh, 2008). Wright et al. (2006) showed that DAC can reduce loads on the low-speed shaft in field tests, compared to a baseline controller.

Exact output regulation (EOR) is a classical linear control systems architecture that is able to track time-varying reference signals while rejecting time-varying disturbances (Saberi et al., 2000). The reference and disturbance signals are assumed to be generated by a known exosystem. EOR feedforwards the exosystem state, where the feedforward gain matrix is obtained as solution of a Sylvester equation, to asymptotically track the reference under the influence of the disturbance. In (Mahdizadeh et al., 2021) EOR was first applied for wind turbines in conjunction with LIDAR, and simulations showed significant fatigue load reduction compared to baseline controller of (Jonkman et al., 2009) and DAC. In (Woolcock et al., 2023) the I&I estimator was used instead of LIDAR, without significantly affecting the control performance.

A notoriously difficult challenge of wind turbine control is the Region 2.5 problem, which is concerned with the transition between below-rated (Region 2) and above-rated (Region 3) operation. This is because the structure of the control problem changes, where in Region 2 blade pitch angles are fixed to a lower saturation and generator torque regulates rotor speed, whereas in Region 3 the roles are reversed, with generator torque being fixed to an upper saturation and blades regulating rotor speed. ROSCO employs a set point smoothing technique that modifies references to the torque and blade pitch controllers, to avoid harmful interaction and ensure that most of the time one saturation is active (Abbas et al., 2022), also see (Schlipf, 2019; Zalkind and Pao, 2019).

The main contributions of this paper are twofold. Firstly, we propose a novel nonlinear wind turbine control design methodology, referred to as nonlinear output regulation (NOR), which solves the Region 2.5 problem directly by design. The main idea is that the closed loop shall follow desired stable dynamics, from which, using a standard nonlinear turbine model, a torque balance equation is derived. In Region 2, this determines the generator torque. If, however, this generator torque exceeds its upper limit, the controller switches to Region 3 and now solves the same equation for the blade pitch angles under saturated generator torque. If this equation fails to admit a solution due to insufficient wind, the controller switches back to Region 2. Hence, NOR creates a seamless transition between Regions 2 and 3, where at any point in time exactly one saturation is active, and, in theory, the closed loop follows the same dynamics across the region switching.

To solve the torque balance equation, NOR uses a wind speed estimate. When using the I&I estimator, we show that the resulting NOR+I&I controller is similar to ROSCO in that linearized closed loops are of the same form, which allows conversion of tuning parameters. We further show that NOR+I&I compensates errors in the model, which alleviates biases in the power coefficient surface. Importantly, NOR's use of a wind speed estimate also allows direct inclusion of LIDAR estimates in a unified manner across operation regions. In particular, we propose to use an average of I&I and LIDAR estimates, which

is the second main contribution of this article. The LIDAR estimate is buffered to be ahead of REWS by the same time as the I&I estimate is lagging behind. By averaging, these delays compensate and an on-time estimate is obtained, which has lower variation than the individual signals because the high frequency noise affecting both estimates has different (stochastically independent) sources.

The performance of NOR was tested with OpenFAST (NREL, 2019) on the IEA 15-MW reference turbine (Gaertner et al., 2020). Full-field wind signals at mean wind speeds between 5 and $20\,\mathrm{m\,s^{-1}}$ were generated with TurbSim (Jonkman, 2006), in accordance with the IEC standard (IEC, 2005). The simulations consider the power generation, the fatigue loads for the tower, blades and main shaft, the blade pitch actuation and rotor speed tracking performance. As baseline controllers for our performance comparisons we use the feedback-only ROSCO (Abbas et al., 2022) as well as the LIDAR-assisted controller (LAC) of Fu et al. (2023) and references therein, which adds a LIDAR-based pitch feedforward to ROSCO, and for which we use the acronym ROSCO+LPFF. Our results show that ROSCO+LPFF, while it expectedly significantly improves tracking performance over feedback-only ROSCO, does not reduce lifetime damage equivalent loads and even increases pitch rate. On the other hand, NOR with the averaged I&I and LIDAR estimate matches ROSCO+LPFF's rotor speed tracking performance, but also significantly reduces fatigue loads and actuator usage. In particular, blade flapwise damage equivalent loads are reduced by 6.7%, which corresponds to a doubling of lifespan, and pitch rate by 36% over ROSCO. The improvements are due to the high quality of the I&I and LIDAR average and NOR's ability to incorporate this information seamlessly in both Regions 2 and 3.

The paper is organized as follows: Section 2 discusses all aspects of the modelling of the 15 MW turbine considered in this study. Section 3 describes the I&I and LIDAR methods used to be used for wind speed estimation, and Section 4 describes the novel NOR and existing ROSCO and ROSCO+LPFF control methodologies to be compared in our performance comparisons. Section 5 describes the Simulink™ simulation environment that was used to test and compare these three control methodologies for their power generation and fatigue loads under a wide range of wind profiles. Section 6 provides the results of our simulations and discusses the relative performance of NOR against ROSCO. Some final remarks are given in the Conclusion, and the Appendix contains some more theoretical analyses and comparisons of NOR and ROSCO.

## 2   Wind turbine modelling

The wind turbine model considered in this work is the IEA 15-MW reference turbine, documented in (Gaertner et al., 2020). It has a variable speed, collective pitch controller; and a low-speed, direct-drive generator. The main parameters are shown in Table 1. Depending on the wind speed, four operating regions are defined. In Region 1, below cut-in speed, there is insufficient wind to power the turbine. In Region 2, which is between cut-in wind speed and rated wind peed, as much power as possible shall be produced. Above rated wind speed, the rated power of $15\,\mathrm{MW}$ can be produced. In Region 3, which is between rated and cut-out wind speed, due to the load capacity of mechanical components and constraints by the generator and grid connection, power output shall be near rated, without overspeeding of the rotor. Operating the turbine in Region 4 would cause excessive loads, so power generation is shut down to protect the turbine.

| | |
|---|---|
| hub height | 150 m |
| rotor radius $R$ | 120 m |
| rated power $P_{rated}$ | 15 MW |
| cut-in wind speed | $3 \, \mathrm{m \, s^{-1}}$ |
| rated wind speed $v_{rated}$ | $10.59 \, \mathrm{m \, s^{-1}}$ |
| cut-out wind speed | $25 \, \mathrm{m \, s^{-1}}$ |
| rated rotor speed $\Omega_{rated}$ | 7.56 rpm |
| minimum rotor speed $\Omega_{min}$ | 5 rpm |
| optimal power coefficient $C_{p,opt}$ | 0.489 |
| optimal tip speed ratio $\lambda_{opt}$ | 9 |
| generator efficiency $\eta_{gen}$ | 95.76% |

**Table 1.** 15-MW reference turbine parameters (Gaertner et al., 2020)

In the following sections we introduce a standard simplified turbine model that will be used to design the NOR controller. Performance simulations will be conducted with the high-fidelity OpenFAST (NREL, 2019) model.

## 2.1 Aerodynamic torque

Let $R$ denote the rotor radius and $\rho$ the air density. The total power of uniform wind moving through the rotor disc, termed *instantaneous power*, is

$$P_{\mathrm{wind}}(v_x) = \frac{1}{2}\rho\pi R^2 v_x^3. \tag{1}$$

where $v_x$ is the magnitude of the horizontal component of the wind velocity vector perpendicular to the rotor plane. The power extracted from the wind by the blades is the *aerodynamic power*, and the ratio of the aerodynamic power to the instantaneous power is given by the *power coefficient* $C_p(\lambda, \theta)$. It depends on the tip speed ratio

$$\lambda = \frac{R\Omega}{v_x}, \tag{2}$$

where $\Omega$ denotes the rotor speed, and the blade pitch angle $\theta$. The power coefficient surface of the 15-MW reference turbine is shown in Figure 1. The maximal power coefficient $C_{\mathrm{p,opt}} = 0.489$ is achieved at optimal tip speed ratio $\lambda_{\mathrm{opt}} = 9$ and $\theta = 0°$. The aerodynamic power and aerodynamic torque are

$$P_a(\Omega, v_x, \theta) = \frac{1}{2}C_p(\lambda, \theta)\rho\pi R^2 v_x^3, \tag{3}$$

$$M_a(\Omega, v_x, \theta) = \frac{1}{2}C_p(\lambda, \theta)\rho\pi R^2 v_x^3/\Omega. \tag{4}$$

The aerodynamic thrust force, which will be relevant for fatigue load reduction, can be modelled similar to the aerodynamic torque as

$$F_a(\Omega, v_x, \theta) = \frac{1}{2}C_t(\lambda, \theta)\rho\pi R^2 v_x^2. \tag{5}$$

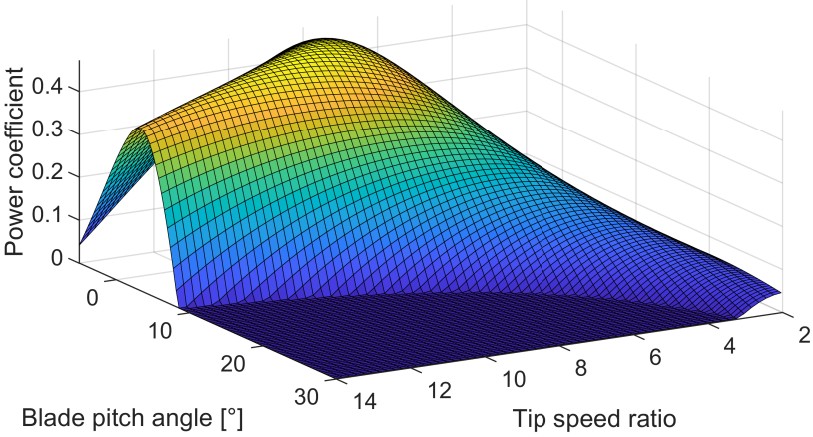

**Figure 1.** Power coefficient surface of the 15-MW reference turbine.

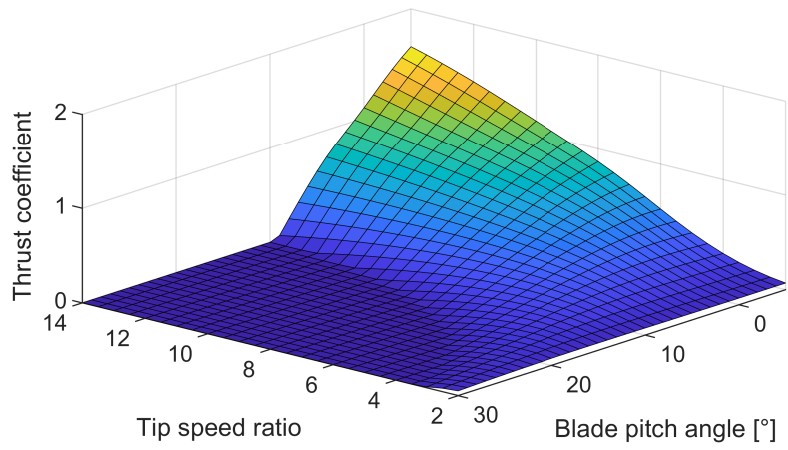

**Figure 2.** Thrust coefficient of the 15-MW reference turbine.

The thrust coefficient surface $C_t(\lambda, \theta)$ is shown in Figure 2. Notice that the thrust coefficient significantly decreases when the blade pitch angle becomes positive. For this reason, the largest thrust forces occur when switching between Regions 2 and 3.

In reality, wind is naturally turbulent, so the previous assumption of $v_x$ as a uniform wind moving through the rotor disc is not accurate. Thus, we consider $v_x$ to be the *rotor-effective wind speed* (REWS) of a turbulent wind field, which is the speed of a uniform wind field that causes the same aerodynamic torque (by (4)) as the turbulent wind field. It is a single wind speed value representing the equivalent horizontal and uniform upstream velocity.

## 2.2 Turbine low dimensional model

Based on Newton's second law of motion, the wind turbine is modelled by the following 1-dimensional nonlinear dynamical model:

$$J\dot{\Omega} = M_a(\Omega, v_x, \theta) - M_g. \tag{6}$$

Here $J = 318,628,138 \, \mathrm{kg \, m^2}$ is the total moment of inertia of the rotating parts, including blades, hub, drivetrain, and generator, obtained from NREL (2021). The mechanical power extracted from the rotor by the generator is

$$P_{\mathrm{mech}} = \Omega M_g, \tag{7}$$

where $M_g$ denotes the generator torque, and the electrical power is modelled as

$$P_{\mathrm{el}} = \eta_{\mathrm{gen}} P_{\mathrm{mech}}, \tag{8}$$

where $\eta_{\mathrm{gen}}$ is the generator efficiency, see Table 1. In order to produce the rated power of $15 \, \mathrm{MW}$, a mechanical power of $15.665 \, \mathrm{MW}$ is required.

## 2.3 Generator and blade pitch actuation

From (6)-(7), we see that the turbine power generation is determined by the rotor speed, and the rotor speed may be controlled by the generator torque and blade pitch angles. The generator torque is controlled via the power electronics in the generator (Gao et al., 2021). The power electronics response is significantly faster than the dynamics of the model (6), as is the blade pitch actuation. Thus, for control design purposes, we disregard these dynamics and treat both $M_g$ and $\theta$ as control inputs that are immediately available for control actuation. In simulations we do assume blade pitch actuation dynamics, however; see Section 5 for details.

## 2.4 Control objective and operating points

In this section we detail the control objectives based on the model (6). The steady-state operating points for $\Omega, M_g, P_{\mathrm{mech}}$ and $\theta$ as functions of the wind speed $v_x$ are shown in Figure 3, and explained in the following. The rotor speed reference $\Omega_{\mathrm{ref}}$ is to be kept between the minimum and maximum/rated speeds $\Omega_{\mathrm{min}}$ and $\Omega_{\mathrm{rated}}$ given in Table 1. The reference $\Omega_{\mathrm{ref}}$ follows the optimal tip speed ratio $\lambda_{\mathrm{opt}}$ whenever these constraints allow, leading to the formula

$$\Omega_{\mathrm{ref}}(v_x) = \max\{\min\{\lambda_{\mathrm{opt}} v_x / R, \Omega_{\mathrm{rated}}\}, \Omega_{\mathrm{min}}\}. \tag{9}$$

Below rated wind speed, the steady-state blade pitch angle $\theta_{\mathrm{ref}}$ is such that the power coefficient is maximized for the current tip speed ratio; note that when the lower saturation $\Omega_{\mathrm{min}}$ is active, the tip speed ratio is below $\lambda_{\mathrm{opt}}$, and highest power coefficient is achieved for positive $\theta$. Above rated wind speed, $\theta_{\mathrm{ref}}$ is such that rated power is achieved, and hence given by the implicit equation

$$P_{\mathrm{rated}} \eta_{\mathrm{gen}}^{-1} = P_a(\Omega_{\mathrm{rated}}, v_x, \theta_{\mathrm{ref}}(v_x)). \tag{10}$$

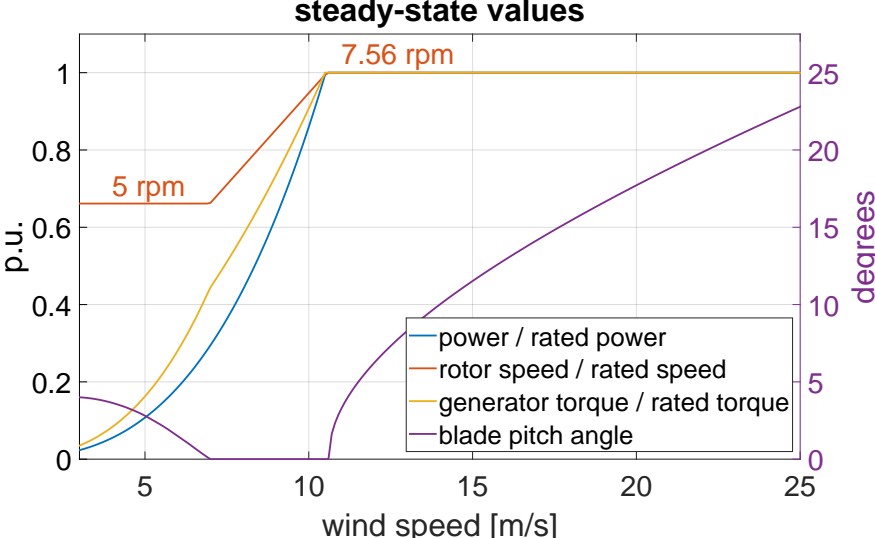

**Figure 3.** Desired equilibrium operating points of (6) without peak shaving.

The power curve in Figure 3 is the aerodynamic power achieved when rotor speed and blade pitch angle follow (9) and (10), i.e.

$$P_{\text{ref}}(v_x) = P_a(\Omega_{\text{ref}}(v_x), v_x, \theta_{\text{ref}}(v_x)). \tag{11}$$

The generator torque curve in Figure 3 is

$$M_{\text{g,ref}}(v_x) = P_{\text{ref}}(v_x)/\Omega_{\text{ref}}(v_x). \tag{12}$$

The main goal of the controller is to track these reference/steady-state values depending on the time-varying REWS $v_x$.

Peak shaving is a technique to reduce aerodynamic thrust force peaks by modifying $\theta_{\text{ref}}$ (Abbas et al., 2022). Figure 4 shows the original $\theta_{\text{ref}}$ and the incurred aerodynamic thrust force $F_a$, computed using (5), as dashed lines. Notice that a sharp peak occurs around rated rotor speed; this is because $F_a$ is proportional to $v_x^2$, but significantly declines as $\theta$ increases, as can be seen in Figure 2. Peak shaving reduces this by already actuating blades slightly below rated wind speed. This reduces power generation. However, thanks to the flatness of the $C_p$-surface (see Figure 1) at low $\theta$, this power sacrifice is relatively small. We limit $F_a$ to $F_{a,\max}$ at around $2\,\text{MN}$. This leads to a minimum pitch schedule $\theta_{\min}(v_x)$, which is obtained by solving

$$F_a(\Omega_{\text{ref}}(v_x), v_x, \theta_{\min}(v_x)) = F_{a,\max} \tag{13}$$

when a solution exists, and equal to the previously defined $\theta_{\text{ref}}$ otherwise. This $\theta_{\min}$, shown in Figure 4, acts as lower saturation for $\theta$ in all our considered controllers.

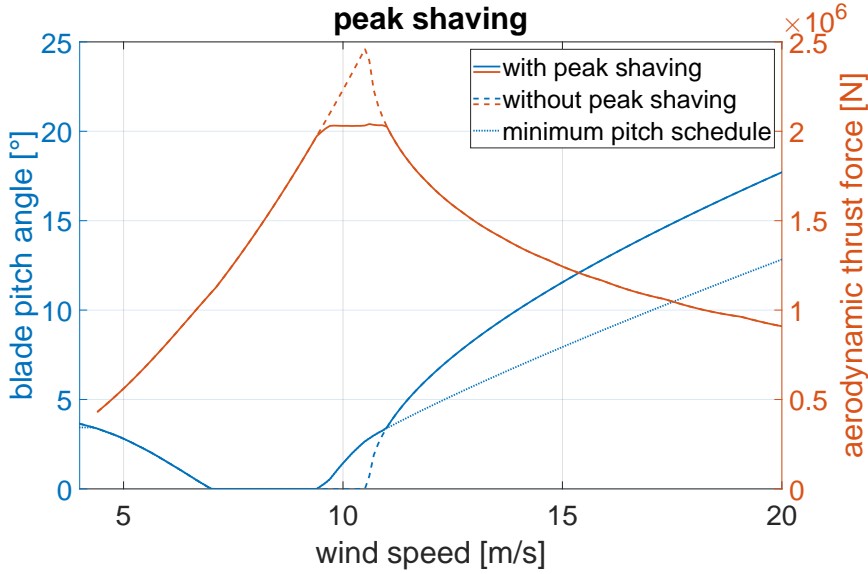

**Figure 4.** Steady-state blade pitch angles (blue) and aerodynamic thrust forces (red) with peak shaving (solid) and without (dashed), and minimum pitch schedule $\theta_{\min}$ (dotted). The dashed blade pitch angle is the same as in Figure 3.

### 2.5  Fatigue loads

Wind turbulence leads to ever-changing stress and strain of the mechanical components, and ultimately material fatigue. Lowering these fatigue loads compared to state-of the art controllers is the main objective of the NOR controller proposed in this work. The four principal fatigue loads, illustrated in Figure 5, and the typical highest displacements that occur under normal operation of the IEA-15MW turbine, are:

1.  tower fore-aft bending moment $M_{yT}$: In wind direction, caused by the aerodynamic thrust force of incoming wind.

2.  tower side-to-side bending moment $M_{xT}$: Sideways, caused by the aerodynamic torque transferred to the tower.

3.  blade flapwise bending moment $M_{yB}$: In wind direction, caused by aerodynamic thrust force.

4.  Main shaft torsion (MST): Torsion of the shaft connecting the rotor with the generator.

The blade edgewise bending moment is another important fatigue load, but it is mostly due to the motion of the blades under gravity. As this cannot be alleviated by any control actuation, we will not consider it in our performance comparisons.

### 2.6  Fatigue load analysis

To estimate the material fatigue, rainflow counting and Miner's rule are used to calculate a damage-equivalent load based on the root moments. As larger load cycles cause disproportionately greater damage, cycle amplitudes are weighted with the Wöhler

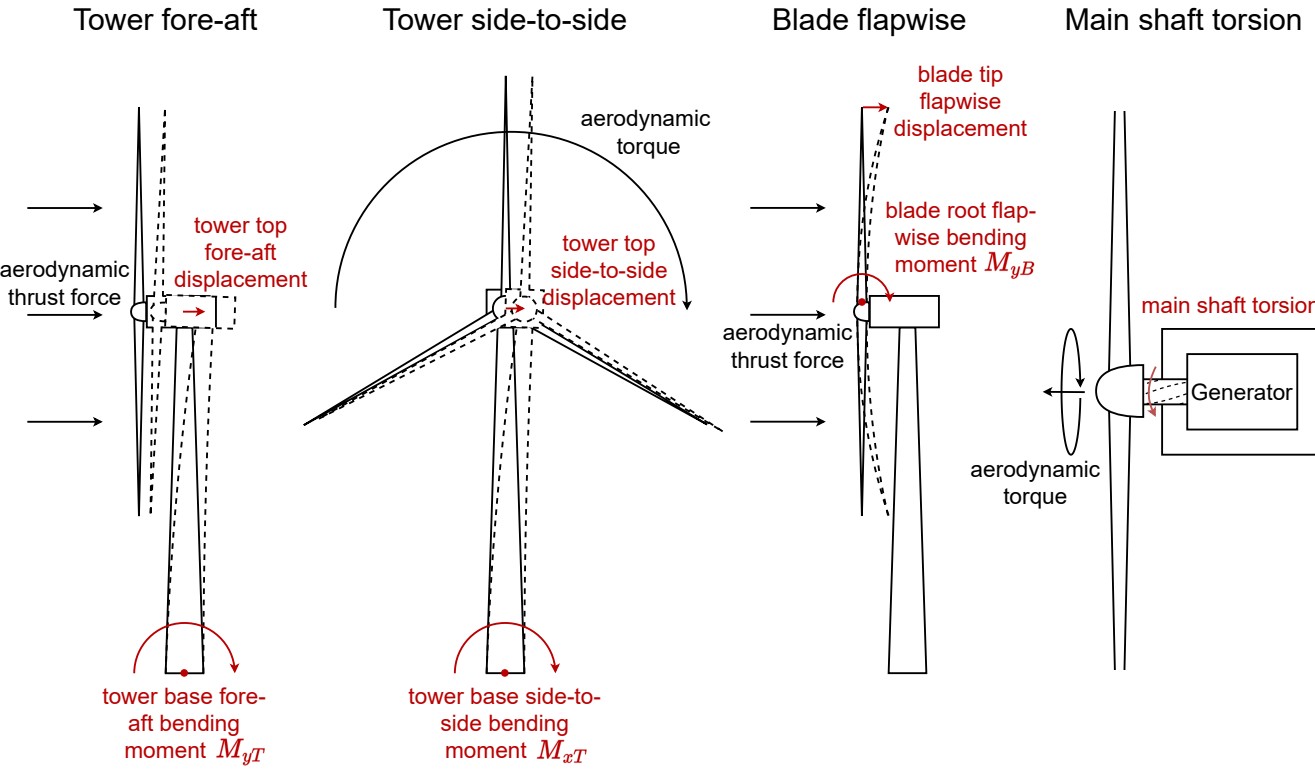

**Figure 5.** Visualizations of principal fatigue loads. Tower fore-aft and blade flapwise bending are mainly a result of aerodynamic thrust force, while tower side-to-side bending and main shaft torsion are mainly caused by aerodynamic torque.

exponent and summed. The damage equivalent load (DEL) is calculated as

$$DEL = \left( \frac{T_{20\,\text{years}}}{T n_{\text{C,ref}}} \sum_{k=1}^{K} n_k A_k^m \right)^{1/m} , \tag{14}$$

where

1. $T_{20\,\text{years}} = 631134720\,\text{s}$ is the nominal lifespan of the turbine, 20 years,

2. $n_{\text{C,ref}} = 2 \times 10^6$ is a reference number of cycles, the value was chosen as in (Schlipf, 2016),

3. $T$ is the time duration of the load history,

4. $K$ is the number of rainflow cycles,

5. $A_k$ is the amplitude of the $k^{\text{th}}$ cycle,

6. $n_k = 0.5$ or $n_k = 1$, depending on whether the $k^{\text{th}}$ cycle is a half cycle or full cycle,

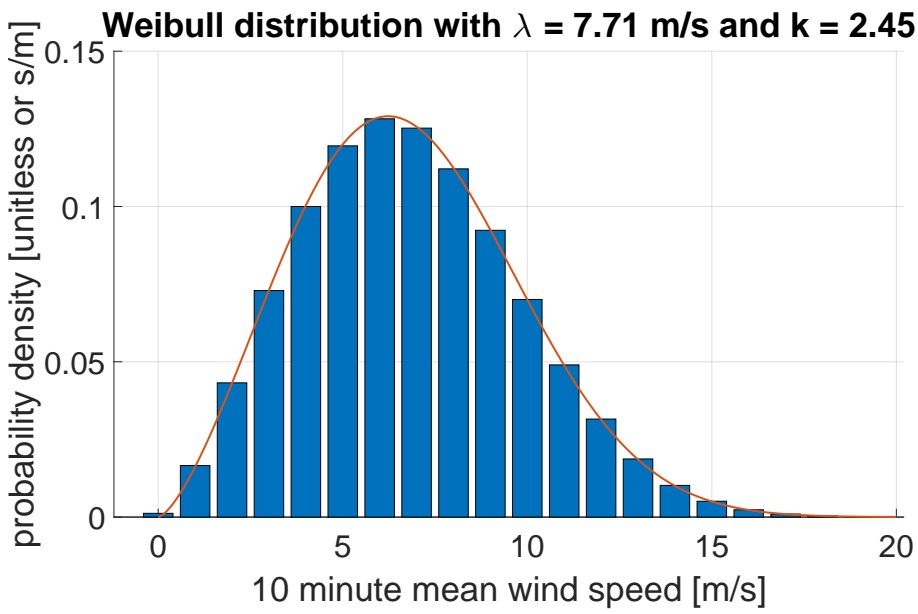

**Figure 6.** Weibull distribution adapted from (Schlipf, 2016, Figure 2.9) based on mean wind speeds recorded in Bremerhaven at a reference height of 102m, with scale parameter $\lambda$ and shape parameter $k$.

7. $m$ is the Wöhler exponent. Empirically found values of $m = 4$ for welded steel (tower material) and $m = 10$ for fibreglass (blade material) are customary (Schlipf, 2016).

The interpretation of (14) is that two million cycles with the DEL as amplitude cause the same damage as the recorded load history, if it is repeated over the nominal lifespan of 20 years.

DELs from simulations at different mean wind speeds are combined according to the Weibull distribution in Figure 6 to obtain a lifetime DEL estimate. This figure shows a histogram obtained from 10-minute-mean wind speeds measured in Bremerhaven, Germany in the winter of 2009 (Schlipf, 2016). The lifetime DEL is a mean of the DELs from the individual simulations, weighted with the probability from the Weibull distribution and the Wöhler exponent.

Due to the high Wöhler exponents, the material fatigue is mostly constituted of a few large cycles, rather than many small cycles. The highest peaks and lowest valleys have a large influence on the DEL. This means that the transition performance between Regions 2 and 3 is crucial, because that is where the highest aerodynamic thrust occurs, as noted in Section 2.5. Wind turbulence naturally leads to a lot of fluctuation in the bending. The aim of our controller design will be to reduce these fluctuations, while still generating maximum power. This will be achieved by smooth yet firm control action to make the peaks and valleys less extreme and avoid excitation of natural frequencies. This will require real-time wind speed estimation, which is discussed in the next section.

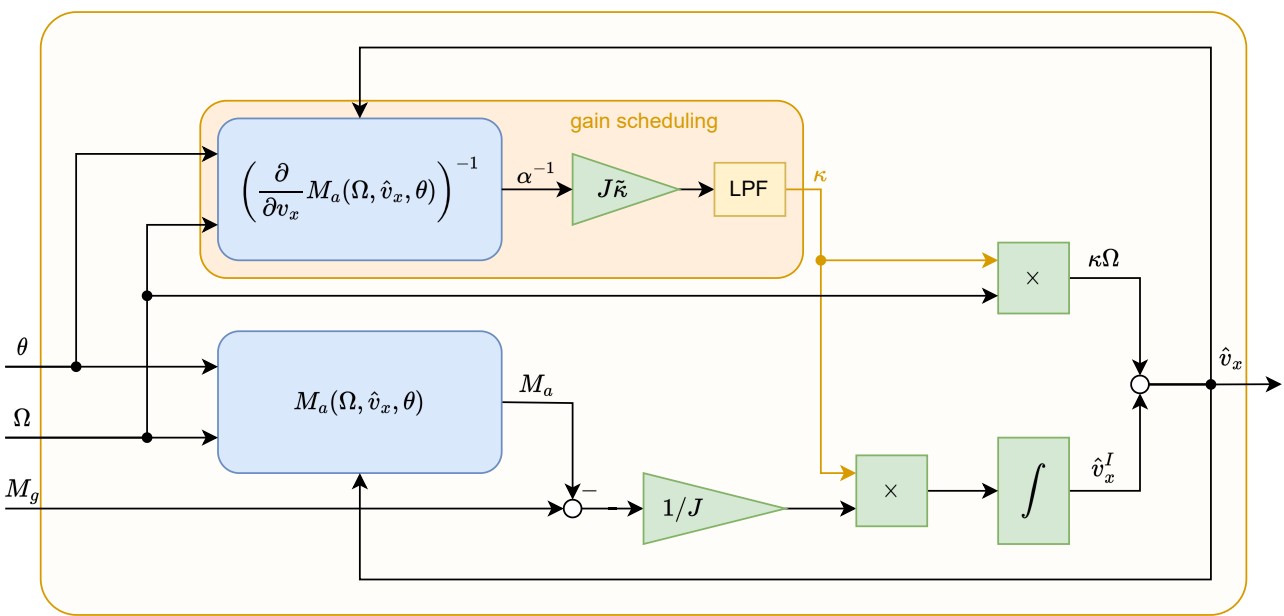

**Figure 7.** Overview of the I&I estimator based on (20) and (21), with gain scheduling described in Section 3.1.1. Inputs are the blade pitch angle $\theta$, rotor speed $\Omega$ and generator torque $M_g$. The output is the wind speed estimate $\hat{v}_x$. Blue boxes mean that a (nonlinear) function is applied, where $M_a(\Omega, v_x, \theta)$ is the aerodynamic torque as in (4). Green triangles are gains, and $\times$ indicates multiplication of signals. LPF means low-pass filter.

## 3   Wind speed estimation

In this section we discuss the two approaches to wind speed estimation that will be employed in our simulations. The first approach uses data available from the turbine's SCADA to compute the rotor effective wind speed (REWS). Many such model-based estimation techniques have been proposed in the wind energy research community. (Soltani et al., 2013) present a comprehensive list of REWS estimators until 2013, and compare them in simulations and field tests. There, the *Immersion and Invariance (I&I)* estimator was found to be among the best performance-wise, while being the simplest to implement. It was recently employed in (Woolcock et al., 2023).

Secondly we briefly describe wind speed estimation obtained from LIDAR measurements.

### 3.1   Immersion & Invariance estimator

Immersion and Invariance was first introduced in (Astolfi and Ortega, 2003) as a tool for stabilization and adaptive control of nonlinear systems, also see (Astolfi et al., 2008). In (Liu et al., 2009) these ideas were adapted for parameter identification of nonlinear systems. The premise here is that a nonlinear system depends on an unknown constant parameter. Based on monotonicity, the I&I estimator converges to that parameter. In (Ortega et al., 2011) and (Ortega et al., 2013) this technique

was applied to wind speed estimation in wind turbines. The wind speed is naturally time-varying, but still treated like the constant unknown parameter. This is suitable, as long as the dynamics of the observer are significantly faster than changes of the REWS.

The general formulation of the I&I estimator based on (Ortega et al., 2013) is:

**Proposition 3.1.** *Consider the system*

$$\dot{x} = F(t) + \Phi(x, \xi) \tag{15}$$

*where $x(t) \in \mathbb{R}$, the function $F(t)$ and the mapping $\Phi : \mathbb{R} \times \mathbb{R} \to \mathbb{R}$ are known, and $\xi \in \mathbb{R}$ is a constant unknown parameter. Assume that there exists a smooth mapping $\beta : \mathbb{R} \to \mathbb{R}$ such that the parametrized mapping*

$$Q_x : \mathbb{R} \to \mathbb{R}, \quad \xi \mapsto \beta'(x)\Phi(x, \xi)$$

*is strictly monotone increasing. Then the I&I estimator*

$$\dot{\hat{\xi}}^I = -\beta'(x)\left(F(t) + \Phi\left(x, \hat{\xi}^I + \beta(x)\right)\right), \tag{16}$$

$$\hat{\xi} = \hat{\xi}^I + \beta(x) \tag{17}$$

*is asymptotically consistent. That is,*

$$\lim_{t \to \infty} \hat{\xi}(t) = \xi \tag{18}$$

*for all $\left(x(0), \hat{\xi}^I(0)\right) \in \mathbb{R} \times \mathbb{R}$ and $F(t)$ such that $\left(x(t), \hat{\xi}(t)\right)$ exist for all $t \geq 0$.*

In (Ortega et al., 2011) and (Ortega et al., 2013) the I&I estimator is applied to the wind turbine dynamics (6), except they only considered Region 2. The generalization to both Region 2 and 3 is immediate. The identification of variables in (15) is:

1. $x := \Omega$ is the rotor speed, $\xi := v_x$ is the rotor-effective wind speed, and $\hat{\xi} := \hat{v}_x$, $\hat{\xi}^I = \hat{v}_x^I$.

2. $F(t) := -M_g(t)/J$ is the acceleration of the rotor from the generator, and $\Phi(x, \xi) := M_a(\Omega, v_x, \theta)/J$ is the acceleration caused by aerodynamic torque.

3. $\beta(x) := \kappa x$ for a design parameter $\kappa > 0$.

A diagram of the I&I estimator is shown in Figure 9. To understand why the estimator works, consider the combined dynamics of the turbine and observer, which are

$$\dot{\Omega} = \frac{1}{J}M_a(\Omega, v_x, \theta) - \frac{1}{J}M_g, \tag{19}$$

$$\dot{\hat{v}}_x^I = \frac{\kappa}{J}\left(M_g - M_a(\Omega, \hat{v}_x, \theta)\right), \tag{20}$$

$$\hat{v}_x = \hat{v}_x^I + \kappa\Omega, \tag{21}$$

leading to

$$\dot{\hat{v}}_x = \dot{\hat{v}}_x^I + \kappa\dot{\Omega} = \frac{\kappa}{J}\left(M_a(\Omega, v_x, \theta) - M_a(\Omega, \hat{v}_x, \theta)\right). \tag{22}$$

Now the need for the monotonicity assumption becomes apparent: If, assuming constant $\Omega$ and $\theta$, the function $M_a(\Omega, v_x, \theta)$ is strictly monotone increasing in $v_x$, then $\hat{v}_x$ converges to $v_x$. The term $v_x^3$ in the aerodynamic torque is monotone increasing. However, $v_x$ also appears in $M_a$ through the tip speed ratio and the power coefficient. For very high or low tip speed ratio the

275 power coefficient drops rapidly, which can cause $M_a(\Omega, v_x, \theta)$ to decrease as $v_x$ increases. In (Ortega et al., 2011) and (Ortega et al., 2013) sufficient conditions for monotonicity of the aerodynamic torque within a certain range are given. In all simulations conducted for this paper the I&I estimator was stable, indicating that under typical operating conditions the monotonicity is fulfilled.

### 3.1.1 Gain-scheduling of I&I estimator

Due to the nonlinearity of the aerodynamic torque, the I&I estimator with constant gain $\kappa$ would have different time constants at different operating points. To counteract this, $\kappa$ is adapted depending on the current wind speed estimate. Let $\tilde{\kappa}$ be the desired I&I cut-off frequency, and let

$$\alpha := \frac{\partial}{\partial v_x} M_a(\Omega, \hat{v}_x, \theta) \tag{23}$$

be the sensitivity of the aerodynamic torque with respect to wind speed at the current rotor speed, wind speed estimate and

285 blade pitch angle. The unfiltered I&I gain is then

$$\kappa_{\text{unfiltered}} = \tilde{\kappa}J\alpha^{-1}. \tag{24}$$

This is passed through a low-pass filter (LPF) with long (slow) time constant to yield $\kappa$, in order to avoid harmful contributions of the gain-scheduling to the estimator dynamics.

### 3.2 LIDAR wind speed estimation

LIDAR uses infrared light and the Doppler effect to measure the horizontal wind speed $v_{\text{x}}$ at a number of evenly distributed points on a circular cross-section of the wind field at the focal distance from the rotor plane (Schlipf et al., 2023). Spatial averaging is applied to the wind speeds at each point to estimate $v_{\text{x}}$ at the focal distance from the turbine blades. Our inclusion of the LIDAR estimate into the controller follows Fu et al. (2023), also see Schlipf (2016). The raw LIDAR signal is passed through a first-order low-pass filter with transfer function

$$G_{LIDAR}(s) = \frac{\omega_{\text{LIDAR}}}{s + \omega_{\text{LIDAR}}} \tag{25}$$

with cutoff frequency $\omega_{\text{LIDAR}}$, which is where the filter gain is -3dB, and where $s$ is the complex frequency. Then, the filtered signal is buffered to synchronize it with the REWS at the rotor. The corresponding time delay is calculated using the LIDAR

lead time $T_{\text{lead}} = \Delta x/\bar{v}_x$, where $\Delta x$ is the LIDAR measurement distance and $\bar{v}_x$ the mean wind speed, the average lidar measurement time (half of the full scan time $T_{\text{scan}}$), and the time delays $T_{\text{filter}}$ and $T_{\text{pitch}}$ caused by the low-pass filter and pitch actuation:

$$T_{\text{buffer}} = \max(T_{\text{lead}} - T_{\text{scan}}/2 - T_{\text{filter}} - T_{\text{pitch}}, 0). \tag{26}$$

The filter time delay can be approximated as $T_{\text{filter}} = 1/\omega_{\text{LIDAR}}$; see (Fu et al., 2023, Equation (30)) for a more detailed approach. See Section 5 for the exact values used in our simulations.

### 3.3 Averaging of I&I and LIDAR estimates

We propose to use an average of I&I and LIDAR estimates, $\hat{v}_x = (\hat{v}_{x,\text{I\&I}} + \hat{v}_{x,\text{LIDAR}})/2$, where the buffer time (26) is modified to

$$T_{\text{buffer}} = \max(T_{\text{lead}} - T_{\text{scan}}/2 - T_{\text{filter}} - 2T_{\text{pitch}} - T_{\text{INI}}, 0), \tag{27}$$

with $T_{\text{INI}} = \tilde{\kappa}$ being the time delay caused by the I&I estimator (see Section 3.1.1). This means that the buffered LIDAR estimate is ahead of REWS by the same time the I&I estimate is lagging behind, therefore their average, accounting for pitch actuation delay, is on time with the REWS at the rotor. The procedure is illustrated in Figure 8. Conceptually, we expect the average estimate, which we denote with the acronym I&I+LIDAR, to be more accurate than the individual signals, as data from different sources are combined. It is reasonable to assume that high frequency noise between I&I and LIDAR signals is stochastically independent, hence the I&I+LIDAR signal has lower variance. This smoother wind speed estimate is expected to cause smoother control actuation, which reduces pitch travel as well as peak loads and ultimately material fatigue. This is verified by simulations in Section 5. Furthermore, we test a weighted average $\hat{v}_x = (1-\alpha)\hat{v}_{x,\text{I\&I}} + \alpha\hat{v}_{x,\text{LIDAR}}$, where $\alpha \in [0,1]$ is the share of LIDAR. The buffer time is then modified to

$$T_{\text{buffer}} = \max(T_{\text{lead}} - T_{\text{scan}}/2 - T_{\text{filter}} - \alpha^{-1}T_{\text{pitch}} - \alpha^{-1}(1-\alpha)T_{\text{INI}}, 0), \tag{28}$$

which is designed to guarantee synchronicity between $\hat{v}_x$ and the actual REWS for linear flanks and coincides with (26) in the all-LIDAR case of $\alpha = 1$.

## 4 Wind turbine control methodologies

In this section we describe the control methodologies that will be compared in our simulations. We first introduce the novel Nonlinear Output Regulation (NOR) controller and its combinations with I&I and LIDAR REWS estimators. As a benchmark for performance comparisons we then briefly describe the ROSCO controller of Abbas et al. (2022), first in its standard feedback-only form and then as LIDAR-assisted control (LAC) with a blade pitch feedforward as in, e.g., Fu et al. (2023), for which we write ROSCO+LPFF. We include ROSCO+LPFF in our simulations to show that NOR+I&I+LIDAR's performance is not only superior to ROSCO (which is expected due to the use of more information), but also an existing LAC method that

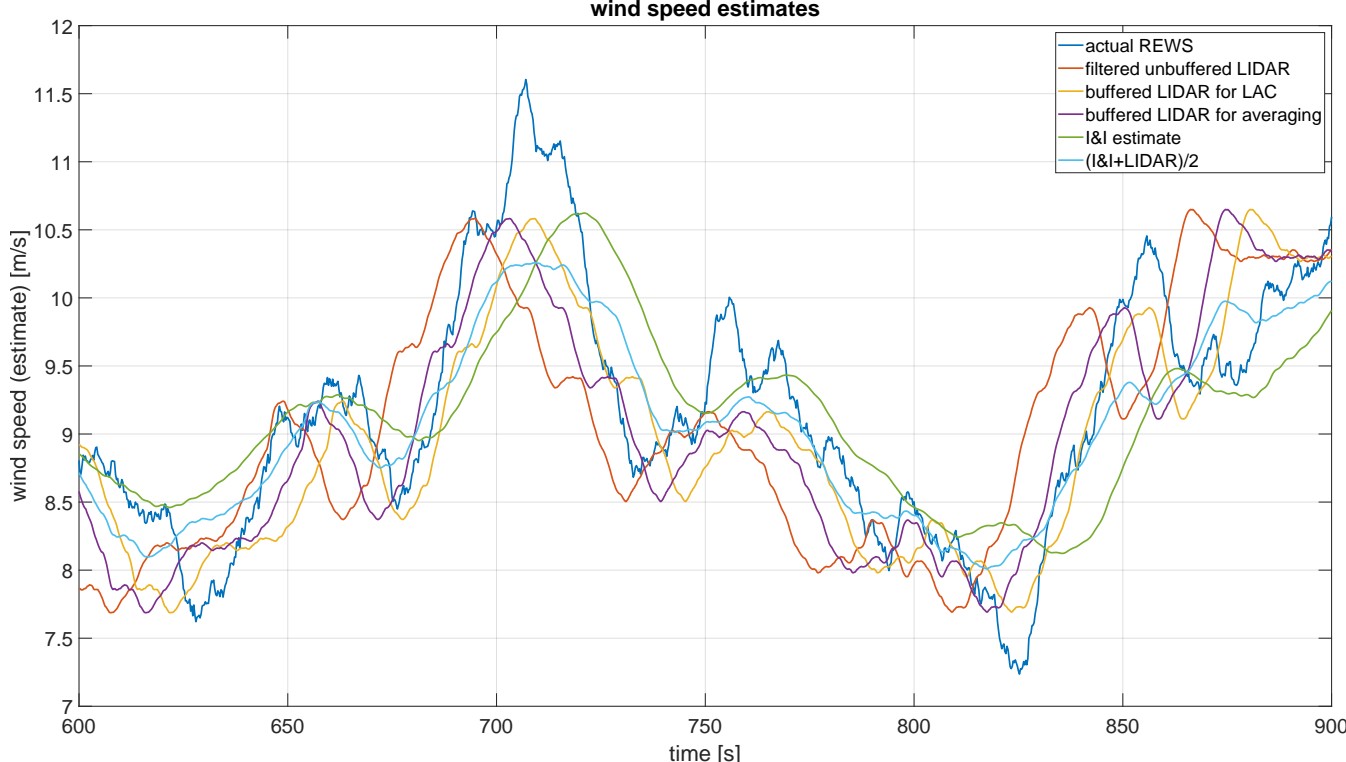

**Figure 8.** Comparison between wind speed estimates; REWS calculated as weighted cubic mean of the wind field (Schlipf, 2016, Section 3.4.2) (blue), LIDAR signal passed through low-pass filter (25) (red), filtered LIDAR signal buffered by (26) to be $T_{\mathrm{pitch}}$ ahead of REWS (yellow), filtered LIDAR signal buffered by (28) to be $2T_{\mathrm{pitch}} + T_{\mathrm{INI}}$ ahead of REWS (purple), I&I estimate (green), and final I&I+LIDAR estimate, i.e., the average of the purple and green estimates (light blue).

uses the same information as NOR+I&I+LIDAR. Finally, we compare the controllers from a conceptual perspective, before the rest of the paper is dedicated to performance comparisons through simulations.

### 4.1 Nonlinear Output Regulation with wind speed estimates

In the following a simple nonlinear wind turbine controller is proposed. The aim is to regulate the turbine rotor speed for Region 2 and 3 power generation, and hence we describe it as Nonlinear Output Regulation (NOR) control. The controller requires a REWS estimate $\hat{v}_x$ that can be the output of any of the estimators in Section 3. Figure 9 shows the closed-loop setup with the I&I estimator, for which we adopt the acronym NOR+I&I. Integration of LIDAR estimates is discussed in Section 4.1.1.

Design is based on the 1-dimensional model for rotor speed dynamics (6), i.e.,

$$J\dot{\Omega} = M_a(\Omega, v_x, \theta) - M_g, \tag{29}$$

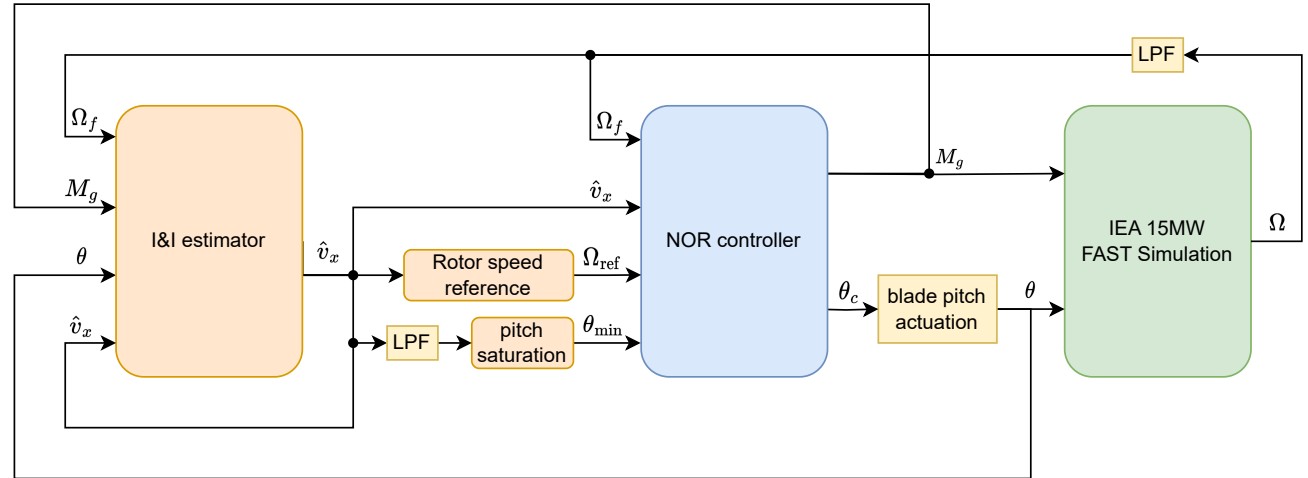

**Figure 9.** Overview of the NOR+I&I closed loop. $\Omega$ and $\Omega_f$ are unfiltered and filtered rotor speed, $M_g$ is generator torque, $\theta_c$ is blade pitch control, $\theta$ is blade pitch angle after the actuation dynamics, and $\hat{v}_x$ is the wind speed estimate. See Figure 7 for the I&I estimator and Algorithm 1 for the NOR controller. The "Rotor speed reference" and "pitch saturation" blocks apply the functions $\Omega_{\text{ref}}$ in (30) and $\theta_{\text{min}}$ in Figure 4, respectively. Yellow blocks are low-pass filters.

where $M_g$ and $\theta$ are control inputs and $v_x$ an exogenous disturbance. The rotor speed reference $\Omega_{\text{ref}}$ is similar to (9), but using the wind speed estimate $\hat{v}_x$ in place of the unknown $v_x$, i.e.,

$$\Omega_{\text{ref}} = \max\left(\min\left(\frac{\lambda_{\text{opt}}\hat{v}_x}{R}, \Omega_{\text{rated}}\right), \Omega_{\text{min}}\right). \tag{30}$$

The idea of the NOR controller is that, if $\hat{v}_x = v_x$, the rotor speed shall follow the desired closed loop dynamics

$$\dot{\Omega} = \mu(\Omega_{\text{ref}} - \Omega), \tag{31}$$

where $\mu > 0$ is a design parameter. Multiplying this with $J$ and equating it with (29) leads to the equation

$$M_a(\Omega, v_x, \theta) - M_g = J\mu(\Omega_{\text{ref}} - \Omega), \tag{32}$$

which the controller must satisfy at all times. As $\theta$ is fixed to $\theta_{\text{min}}$ in Region 2 and $M_g$ is constrained in Region 3, NOR chooses
the remaining free control variable (that is, $M_g$ in Region 2 and $\theta$ in Region 3) depending on $\Omega$ and $\hat{v}_x$ such that (32) holds, and switches between the regions when required. This leads to the control law given in Algorithm 1, which is further explained in the following. For this explanation we disregard the rotor speed low-pass filter and blade pitch actuation dynamics, i.e., assume $\Omega_f = \Omega$ and $\theta = \theta_c$.

    The parameter $\varphi \in [0, 1]$ indicates whether the controller in Region 3 is designed to track rated torque, rated power or a
combination of these. By (35), $\varphi = 1$ leads to constant torque $M_g = M_{\text{rated}}$, and $\varphi = 0$ leads to constant power by $M_g = P_{\text{rated}}/\Omega$. In all subsequent simulations $\varphi = 1$ is used. The minimum pitch schedule $\theta_{\text{min}}$ is as in Figure 4.

---

**Algorithm 1** Nonlinear output regulation (NOR) controller

---

**Inputs:** wind speed estimate $\hat{v}_x$, low-pass filtered rotor speed $\Omega_f$

**Outputs:** generator torque $M_g$, blade pitch angle control $\theta_c$

    **if** $M_a(\Omega_f, \hat{v}_x, \theta_{\min}(\hat{v}_x)) + J\mu(\Omega_f - \Omega_{\text{ref}}) \leq \varphi M_{\text{rated}} + \frac{(1-\varphi)P_{\text{rated}}}{\Omega_f}$ **then**

        (Region 2 / below-rated operation)

$$\theta_c \leftarrow \theta_{\min}(\hat{v}_x) \tag{33}$$

$$M_g \leftarrow M_a(\Omega_f, \hat{v}_x, \theta_{\min}(\hat{v}_x)) + J\mu(\Omega_f - \Omega_{\text{ref}}) \tag{34}$$

    **else**

        (Region 3 / above-rated operation)

$$M_g \leftarrow \varphi M_{\text{rated}} + (1-\varphi)\frac{P_{\text{rated}}}{\Omega_f} \tag{35}$$

        solve $M_g = M_a(\Omega_f, \hat{v}_x, \theta_c) + J\mu(\Omega_f - \Omega_{\text{ref}})$ for $\theta_c$       (36)

    **end if**

---

This controller regulates rotor speed in Region 2 using the generator torque, by compensating aerodynamic torque and adding a feedback term based on the desired closed loop dynamics (see (34), note the similarity to (32)). If this generator torque control (34) were to exceed rated torque (if $\varphi = 1$) or rated power (if $\varphi = 0$), i.e. the inequality in the if-condition is not satisfied, then the else-branch representing the Region 3 control applies. Then, equation (36) always has a unique solution $\theta$ that is at least $\theta_{\min}(\hat{v}_x)$, as detailed in Remark 4.1 below. By substituting in $\theta$ and $M_g$ (from either region) into (29), it can be seen that the closed loop always follows (31), if the wind speed estimate is perfectly accurate, i.e. $\hat{v}_x = v_x$, as intended.

**Remark 4.1.** *Existence and uniqueness of a solution $\theta > \theta_{min}(\hat{v}_x)$ of (36) is due to the intermediate value theorem and monotonicity. Indeed, $M_a(\Omega, \hat{v}_x, \theta_{min}(\hat{v}_x)) - M_g > J\mu(\Omega_{ref} - \Omega)$ in Region 3, which means that $\theta_{min}$ would give higher aerodynamic torque than (36) demands. For high blade pitch angles the power coefficient becomes 0 or even negative. In between, due to continuity of the $C_p$ surface and the intermediate value theorem, a solution exists. This solution is unique since, for any relevant tip speed ratio, the power coefficient is monotone decreasing in $\theta$ when $\theta > \theta_{min}$. This is not directly obvious; for example, for a tip speed ratio of 4, the maximum power coefficient is attained at around $5.8°$, meaning that there are potentially two solutions for $\theta_c$ above and below that. However, $\theta_{min}$ from peak shaving is always way above this threshold (then around $15°$), thus restricting it to the higher, then unique, pitch angle. If, despite this, the solution is not unique at times (perhaps due to unusually high or low rotor speeds), then the larger of the two solutions should be chosen.*

The NOR controller performs a smooth transition between Regions 2 and 3, meaning that $\theta$ and $M_g$ do not jump. Indeed, at the time of region switching it holds that

$$M_a(\Omega, \hat{v}_x, \theta_{\min}(\hat{v}_x)) + J\mu(\Omega - \Omega_{\text{ref}}) = \varphi M_{\text{rated}} + (1-\varphi)\frac{P_{\text{rated}}}{\Omega}, \tag{37}$$

under the assumption that all the signals are continuous in time. The torque controls (34) and (35) in Regions 2 and 3 are the left and right-hand side of that equation and therefore coincide. Furthermore, the solution of (36) is then $\theta_{\min}(\hat{v}_x)$, and indeed equal to (33).

NOR+I&I's design parameters are the gains $\tilde{\kappa}$ and $\mu$. Increasing them improves rotor speed tracking at the cost of higher actuator usage. Different values may be chosen depending on mean wind speed, or gain-scheduling may be applied. Furthermore, the maximum aerodynamic thrust in (13) that generates $\theta_{\min}$ for peak shaving can be adjusted to trade off thrust related DELs (i.e., tower fore-aft and blade flapwise DELs) and power sacrifice.

NOR+I&I compensates model errors in the $C_p$-surface (that NOR heavily relies upon) without the need of an integrator in the controller. Intuitively, this is because, since I&I and NOR use the same $C_p$-surface, any over- or underestimation of REWS by the I&I estimator has the opposite effect in the NOR controller, thus cancelling out errors. In fact, suppose the power coefficients are assumed too high, then the I&I estimator overestimates the aerodynamic torque caused by any given wind speed. This leads it to underestimate the wind speed based on the actual aerodynamic torque the rotor experiences. The NOR controller now works with an underestimated wind speed estimate and an overestimated $C_p$ surface. These two effects compensate in the aerodynamic torque formula (4). The NOR controller then chooses $M_g$ in (34) and $\theta_c$ in (36) to balance out the actual aerodynamic torque and thereby avoids persistent tracking errors. This is mathematically proven in Appendix A.

**Remark 4.2.** *The NOR controller is a nonlinear and simplified version of the exact output regulation (EOR) controller of Mahdizadeh et al. (2021); Woolcock et al. (2023), which is a linear control methodology that constructs exosystems to model low-order approximate wind speed dynamics and uses them to generate pitch and torque feedforwards which, in theory, regulate rotor speed under changing wind speed and account for shaft torsion and blade pitch actuation dynamics. We found, however, that neglecting shaft torsion and blade pitch actuation dynamics does not lower performance. This simplification allows the nonlinear approach of NOR which foregoes the need for linearization, gain scheduling and exosystems, and also enables the smooth switching between Regions 2 and 3. The NOR controller can be seen as a to the best of our knowledge novel version of disturbance accommodation control.*

### 4.1.1 LIDAR inclusion into the NOR controller

When LIDAR or I&I+LIDAR is used instead of just the I&I estimator, a mean correction should be applied to the (filtered and buffered) LIDAR estimate as shown in Figure 10. Otherwise, errors in the $C_p$-surface lead to persistent discrepancies in rotor speed and power tracking. The mean correction ensures that the combination of estimator and controller benefits from the error correction phenomenon mentioned above for NOR+I&I and detailed in Appendix A. To achieve this mean correction, the difference between I&I and LIDAR estimates is passed through a slow low-pass filter (where the time constant should be at least a minute) and then added back onto LIDAR. The resulting signal has the high frequency content of LIDAR and low frequency content of the I&I estimate.

**Remark 4.3.** *When we simulated NOR+I&I+LIDAR without this mean correction, a significant discrepancy of around 5 to 10% in Region 3 average power generation occurred, indicating that the mean correction is essential. The reason for the*

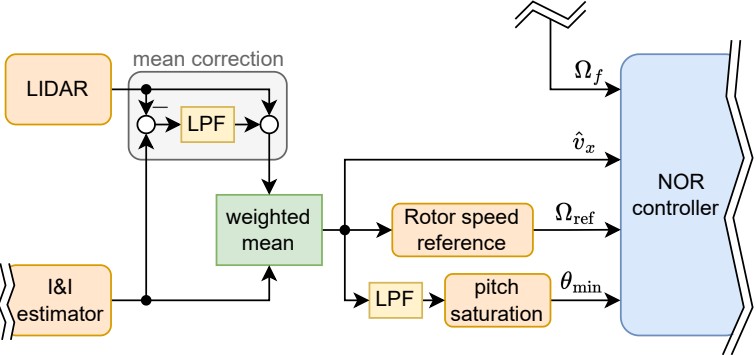

**Figure 10.** Section of Figure 9, with the addition of LIDAR.

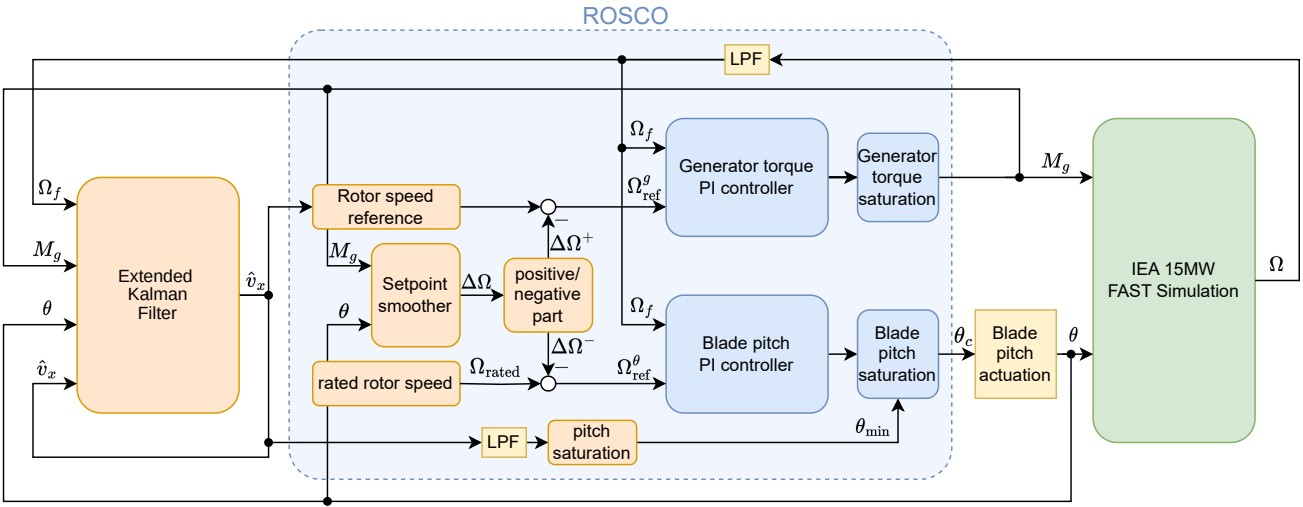

**Figure 11.** Overview of the ROSCO simulation setup. Rotor speed regulation is achieved by two SISO PI controllers, where saturations and set point smoothing avoid conflict between them. Yellow blocks are low-pass filters.

*discrepancy was likely due to biases in the $C_p$ surface. Note that our simulations use the high fidelity OpenFAST model, so a difference between the modelled $C_p$ surface and the actual behaviour in simulations is expected. In Section 6 we will see*
*that NOR+I&I(+LIDAR) in our simulations very accurately tracks rated rotor speed and rated power in Region 3 on average without a constant offset. This indicates that the bias correction for the $C_p$ surface detailed in Appendix A and the mean correction introduced above indeed work in practice.*

## 4.2 Reference Open-Source baseline controller (ROSCO)

To compare the performance of the NOR controller, we shall use ROSCO, the state-of-the-art modular reference wind turbine controller that was shown to perform comparably to, or better than, previous reference controllers in (Abbas et al., 2022). Figure 11 shows an overview of the ROSCO setup used in this paper. The components are explained in the following, see Section 5 for tuning.

**Wind speed estimator.** A REWS estimate is used to determine rotor speed reference and minimum blade pitch angles. In our simulations we use ROSCO's default Extended Kalman Filter.

**Torque and blade pitch PI controllers.** For the generator torque and blade pitch angle controls, independent single-input single-output PI controllers, i.e.

$$M_g = K_P^g(\Omega_{\text{ref}}^g - \Omega_f) + K_I^g \int (\Omega_{\text{ref}}^g - \Omega_f)\, \mathrm{d}t, \tag{38}$$

$$\theta_c = K_P^\theta(\Omega_{\text{ref}}^\theta - \Omega_f) + K_I^\theta \int (\Omega_{\text{ref}}^\theta - \Omega_f)\, \mathrm{d}t, \tag{39}$$

are used. To account for nonlinearity of the aerodynamic torque, the blade pitch controller is gain-scheduled. Details on that can be found in Section B. The generator torque control is saturated at rated torque (this usually occurs in Region 3). The blade pitch angle control is saturated to be at least a certain minimum blade pitch angle (this usually occurs in Region 2 and around rated wind speed), details are further below.

**Reference signal and set point smoothing.** The rotor speed reference $\Omega_{\text{ref}}^g$ for the generator torque controller is based on (9), i.e., optimal tip speed ratio saturated between minimum and rated rotor speed. The rotor speed reference $\Omega_{\text{ref}}^\theta$ for the blade pitch controller is based on rated rotor speed. In order to ensure proper region switching, these values are modified by a set point smoothing technique. A correction term

$$\Delta\Omega = \left[ \left( \frac{\theta - \theta_{\min}}{\theta_{\max}} \right) k_{\text{vs}} - \left( \frac{M_{\text{g,rated}} - M_g}{M_{\text{g,rated}}} \right) k_{\text{pc}} \right] \Omega_{\text{rated}} \tag{40}$$

with design parameters $k_{\text{vs}}$ and $k_{\text{pc}}$ is computed. If it is positive, it is subtracted from the torque control reference, and if negative, subtracted from the pitch control reference instead. In view of Figure 11,

$$\Delta\Omega^+ = \max\{\Delta\Omega, 0\}, \quad \Omega_{\text{ref}}^g = \Omega_{\text{ref}} - \Delta\Omega^+,$$

$$\Delta\Omega^- = \min\{\Delta\Omega, 0\}, \quad \Omega_{\text{ref}}^\theta = \Omega_{\text{rated}} - \Delta\Omega^-.$$

Set point smoothing ensures that most of the time one of the controllers is saturated.

**Saturations.** Generator torque is saturated with rated torque as upper limit. Blade pitch commands are saturated with $\theta_{\min}$ as lower limit (see Figure 4) for power maximization at low wind speeds and peak shaving.

**Additional filters.** Generator speed, as well as the wind speed estimate going into the pitch saturation, are low-pass filtered, in order to lessen the effect of measurement errors and reduce control actuation.

### 4.3 ROSCO with LIDAR-assisted feedforward pitch control (ROSCO+LPFF)

Augmenting conventional feedback controllers like ROSCO with LIDAR-based pitch feedforward, then referred to as LIDAR assisted control (LAC), can significantly improve performance, see, e.g., Schlipf (2016); Fu et al. (2023). Following the design of Fu et al. (2023) and references therein, a feedforward blade pitch angle is computed as

$$\theta_{\text{FF}} = \theta_{\text{ref}}(\hat{v}_{x,\text{LIDAR}}), \tag{41}$$

where $\theta_{\text{ref}}$ is the steady-state pitch curve defined in (10), also see Figure 3, and $\hat{v}_{x,\text{LIDAR}}$ is the REWS estimate obtained from LIDAR as in Section 3.2. For reasons detailed in Schlipf (2016), $\theta_{\text{FF}}$, instead of directly being added to the feedback terms in (39), is differentiated and added before the integrator, leading to the following pitch control law to replace (39):

$$\theta_c = K_P^\theta (\Omega_{\text{ref}}^\theta - \Omega_f) + \int \left( K_I^\theta (\Omega_{\text{ref}}^\theta - \Omega_f) + \dot{\theta}_{\text{FF}} \right) \text{d}t. \tag{42}$$

All other components of the ROSCO+LPFF controller in our study are the same as ROSCO.

### 4.4 Comparison between NOR+I&I and ROSCO

We briefly compare NOR+I&I and ROSCO from a conceptual point of view. Note that NOR+I&I, even though it uses $\hat{v}_x$ as feedforward, is still a feedback controller (like ROSCO), because the I&I estimator computes $\hat{v}_x$ from plant outputs. The first order plant dynamics (29), NOR (which uses a static control law with no internal states) and the first order I&I dynamics (22) lead to a second-order combined closed-loop, similar to the second-order dynamics that ROSCO is designed for. In Appendix B we provide a theoretical comparison between these two closed loops via linearization. This enables conversion of the tuning choices for the design parameters, in particular the proportional and integral gains of ROSCO to $\mu$ and $\tilde{\kappa}$ of NOR+I&I and conversely. We use this procedure to tune NOR+I&I based on ROSCO's tuning, see Section 5 for details. This analysis identifying the linearized closed loops only works in Regions 2 and Region 3 in isolation, and does not take into account the region switching. Here lies one of the differences between NOR and ROSCO; while ROSCO employs set-point smoothing to avoid harmful interference between its two SISO loops, NOR by design transitions seamlessly between the regions. This, together with the direct inclusion of the wind speed estimate $\hat{v}_x$, allows effective use of LIDAR preview information in both regions. In particular, the region switching is enhanced by LIDAR, whereas the ROSCO+LPFF in Section 4.3 only uses LIDAR feedforward in Region 3. In fact, our simulation results in Section 6 show that NOR+I&I+LIDAR provides superior performance to ROSCO+LPFF particularly when switching regions. Finally, NOR permits direct inclusion of peak shaving and pitch actuation at low wind speeds into the control design through $\theta_{\text{min}}$, whereas ROSCO applies it as a saturation after the blade pitch controller. The difference is then that the torque controller of NOR is "informed" of these blade pitch changes, whereas ROSCO's torque controller is not.

 **5   Turbine Simulation Environment and Tuning**

Here we describe the various components of our simulation environment for testing the performance of the ROSCO and NOR controllers.

**Turbine and wind simulation.** We used OpenFAST (NREL, 2019) as our turbine simulator on the IEA 15-MW reference turbine in the fixed-bottom monopile configuration (Gaertner et al., 2020). Full-field wind signals of length 120 minutes with
mean wind speeds between 5 and $20\,\mathrm{m\,s^{-1}}$ (at hub height) were generated with TurbSim using the parameters in Table 2.

**LIDAR simulation.** We employed the simulation toolbox given in Schlipf et al. (2023); Guo et al. (2023) using the nacelle-mounted Molas NL400 LIDAR system with 4 measuring points. We adopted the measurement distance and opening angle that were found in Fu et al. (2023) to maximize coherence between the processed LIDAR estimate and actual REWS for this type of LIDAR system, shown in Table 3. Furthermore, we adopt the same parameters for the filtering and buffering as in Fu et al.
(2023). For simplicity we adopted Taylor's frozen wind hypothesis, meaning that the wind field was assumed not to evolve between the focal point and the blades, and hence the wind evolution component of the toolbox was not used. This hypothesis is appropriate for relatively flat terrain where geological features do not interact with the air flow between the measurement point and blades.  Blade blockage of the LIDAR beam is neglected.

**ROSCO implementation.** ROSCO is applied with peak-shaving and set point smoothing using the default parameters for
the IEA 15MW reference turbine, shown in Table 4. We simulate both the feedback-only form of ROSCO introduced in Section 4.2 and the ROSCO+LPFF modification introduced in Section 4.3.

**NOR tuning.** We choose $\tilde{\kappa}$ and $\eta$ depending on mean wind speed. The NOR controller is tuned based on the tuning choices of ROSCO according to equations (B19) and (B20) in Appendix B, which results in NOR+I&I and ROSCO having the same linearized closed loop dynamics. This gives  $\tilde{\kappa} = \eta = 0.12\,\mathrm{s^{-1}}$ for below-rated mean wind speed, and $\tilde{\kappa} = \eta = 0.2\,\mathrm{s^{-1}}$ above
rated. To create a transition, we keep $\tilde{\kappa}$ and $\eta$ constant in each simulation based on the mean wind speed, with values of $0.12$ up to $8.59\,\mathrm{m\,s^{-1}}$, and then increasing linearly to $0.2$ from $12.59\,\mathrm{m\,s^{-1}}$. Peak shaving is built into the NOR controller as outlined in Section 4.1.

**Additional filters.** To model blade pitch actuation, a second order low-pass filter with undamped natural frequency $\pi/2\,\mathrm{rad\,s^{-1}}$ and damping factor $0.7$ is included between the blade pitch angle output of the controllers and the blade pitch angle input of
OpenFAST. The following low-pass filters, as indicated in Figure 9, and with parameters given in Table 5, are used: In ROSCO, the rotor speed measurement is by default passed through a second-order low-pass filter. The wind speed estimate, that minimum blade pitch angles are computed with, is passed through a first-order low-pass filter. For the sake of comparability, the same filters are used for NOR. Finally, the variable gain of the gain-scheduled I&I estimator is filtered with a first-order low-pass filter, to avoid harmful effects of the gain-scheduling on the dynamics.

**Performance metrics.** For each controller and each of the 16 mean wind speeds, eight performance metrics are computed as follows from simulation data in the time interval between $t_{\mathrm{start}} = 300\,\mathrm{s}$ and $t_{\mathrm{end}} = 7200\,\mathrm{s}$:

   – The average power generation,

| Duration | 2 h |
|---|---|
| Turbulence class | B |
| Turbulence model | IECKAI |
| Power law exponent | default |
| Time step | 0.25 s |
| number of grid points in vertical direction | 33 |
| number of grid points in horizontal direction | 33 |
| hub height | 150 m |
| grid height | 250 m |
| grid width | 250 m |

**Table 2.** TurbSim parameters

| Measurement distance | 220 m |
|---|---|
| Opening angle | 15° |
| scan time $T_{\text{scan}}$ | 0.25 s |
| LIDAR low-pass filter $\omega_{\text{LIDAR}}$ | 0.1467 rad/s |
| LIDAR low-pass filter delay $T_{\text{filter}}$ | 6.7067 s |
| pitch actuation delay $T_{\text{pitch}}$ | 0.9 s |

**Table 3.** Parameters of the LIDAR measuring system and data processing

| Variable speed closed-loop frequency $\omega_{\text{des}}^g$ | $0.12\,\text{rad s}^{-1}$ |
|---|---|
| Variable speed closed-loop damping $\zeta_{\text{des}}^g$ | 1 |
| Pitch control closed-loop frequency $\omega_{\text{des}}^\theta$ | $0.2\,\text{rad s}^{-1}$ |
| Pitch control closed-loop damping $\zeta_{\text{des}}^\theta$ | 1 |
| Set point smoothing variable speed gain $k_{\text{vs}}$ | 1 |
| Set point smoothing pitch control gain $k_{\text{pc}}$ | $10^{-3}$ |
| Set point smoothing max. blade pitch angle $\theta_{\text{max}}$ | 30° |

**Table 4.** ROSCO design parameters

| Generator speed 2nd-order LPF frequency | $1.008\,\text{rad s}^{-1}$ |
|---|---|
| Generator speed 2nd-order LPF damping | 0.7 |
| Wind speed 1st-order LPF for pitch saturation | $0.21\,\text{rad s}^{-1}$ |
| I&I variable gain 1st-order LPF | $0.033\,\text{rad s}^{-1}$ |

**Table 5.** Additional low-pass filter parameters. The frequencies refer to the -3dB cutoff frequencies.

– four metrics relating to the turbine fatigue loads, which are the tower fore-aft and side-to-side DELs based on root moments, with Wöhler exponent 4; the average of the three blade flapwise DELs based on root moments, with Wöhler exponent 10, and the main shaft torsion DEL based on rotor torque, with Wöhler exponent 4.

– The average pitch rate as a measure for pitch actuator wear,

$$\frac{1}{t_{\text{end}} - t_{\text{start}}} \int\limits_{t_{\text{start}}}^{t_{\text{end}}} |\dot{\theta}(t)| \mathrm{d}t,$$

– The root mean square error between rotor speed $\Omega(t)$ and $\Omega_{\text{rated}}$, but only accounting for above-rated wind speeds, i.e.,

$$\frac{1}{t_{\text{end}} - t_{\text{start}}} \sqrt{\int\limits_{t_{\text{start}}}^{t_{\text{end}}} \alpha(v_x(t))(\Omega(t) - \Omega_{\text{ref}}(\hat{v}_x(t)))^2 \, \mathrm{d}t},$$

where $\alpha(v_x) = 1$ whenever $v_x > v_{\text{rated}}$ and 0 otherwise, and $v_x$ being calculated as weighted cubic mean of the wind field at the rotor (Schlipf, 2016, Section 3.4.2).

– The maximum rotor speed.

With the exception of the average power generation, a smaller measure indicates superior performance in all cases.

For the DELs, we computed lifetime weighted means across all wind speeds according to Wöhler exponent and a Weibull distribution as described in Section 2.6. Because the Weibull distribution in Figure 6 models wind speeds at a reference height of $102$m, we scale up the distribution according to the power law with exponent $0.2$ to a reference height of $150$m, resulting in a new scale parameter of $8.33\,\mathrm{m\,s}^{-1}$. For average power and pitch rate we calculate the arithmetic mean weighted with Weibull distribution, for RMS error the quadratic mean weighted with Weibull distribution, and for maximum rotor speed the maximum across all wind speeds.

## 6  Performance results and comparisons

**Overview.** The results of our turbine simulations comparing feedback-only ROSCO, ROSCO+LPFF and NOR+I&I+LIDAR are shown for individual mean wind speeds in Figure 12 and averaged across all wind speeds in Table 6. See Figure 13 for a 3-minute time series that illustrates how NOR+I&I+LIDAR achieves fatigue load reduction. Figure 14 illustrates the effects of different weightings between I&I and LIDAR. The results are discussed in the following.

**Lifetime fatigue loads.** Figure 12 shows that NOR+I&I+LIDAR achieves significant reductions in tower fore-aft DELs for above-rated mean wind speeds compared to both ROSCO and ROSCO+LPFF. However, highest tower DELs occur at low wind speeds due to resonance between the monopile of the IEA15MW model, which has a natural frequency of around $1.5\,\mathrm{rad/s}$, and the 3P frequency of the rotor at $5\,\mathrm{rpm}$. Particularly the side-to-side bending is very lightly damped, leading to large random variation in the DEL across simulations, which explains the spike of NOR+I&I+LIDAR at $6\,\mathrm{m/s}$. For a turbine model where

resonance between monopile and 3P does not occur, NOR+I&I+LIDAR would achieve more significant tower fore-aft lifetime DEL reduction compared to ROSCO and ROSCO+LPFF than reported in Table 6. NOR+I&I+LIDAR achieves a significant blade flapwise lifetime DEL reduction of $6.68\%$ over ROSCO, whereas ROSCO+LPFF slightly increases it, mostly due to its performance near rated wind speed. Based on formula (14) with Wöhler exponent 10, a DEL reduction of $6.68\%$ corresponds to an increase in lifespan of $99.6\%$, i.e., almost doubling it.

**Lifetime acuator usage.** While the LIDAR feedfoward in ROSCO+LPFF causes increased pitch rate compared to ROSCO, NOR+I&I+LIDAR achieves a substantial reduction across all wind speeds. Compared to ROSCO+LPFF, NOR+I&I+LIDAR reduces pitch rate by more than 40%. This significantly reduces wear on the pitch actuation system. The reduction in pitch rate does not come at the cost of power generation and rotor speed tracking performance compared to ROSCO+LPFF. The pitch rate reduction is mainly due to the averaging of I&I and LIDAR estimates, which creates a smoother feedforward signal with less high frequency content. Figure 13 confirms that the blade pitch commands of NOR+I&I+LIDAR are much more steady and less oscillatory than those of ROSCO and ROSCO+LPFF.

**Results at 18 m/s.** Table 7 compares the DELs at a mean wind speed of $18\,\mathrm{m/s}$, including the results reported in Fu et al. (2023) for ROSCO+LPFF using a 4 beam continuous wave LIDAR with the same tuning and turbine model as in our study. Our study has replicated the performance numbers in Fu et al. (2023) quite closely. They are not identical due to differences in the wind field, where Fu et al. (2023) consider an evolving field whereas we assume Taylor's frozen wind hypothesis, and different simulation runtimes. NOR+I&I+LIDAR achieves about twice the reduction in tower fore-aft and blade flapwise DELs of ROSCO+LPFF over ROSCO at this wind speed, while being only slightly worse on rotor speed tracking. This shows that NOR+I&I+LIDAR's improvements over ROSCO+LPFF are not only due to the use of LIDAR in both Regions rather than just Region 3; even in Region 3 in isolation NOR+I&I+LIDAR is superior.

**Weighting of I&I and LIDAR.** Figure 14 confirms that the averaged I&I and LIDAR estimate yields much better performance than when only the individual I&I or LIDAR signals are used for NOR. Lowest rotor speed RMS error is achieved when weighting I&I slightly more, whereas blade flapwise DEL benefits from higher LIDAR contribution. The fact that tower-foreaft DEL is lowest at full LIDAR is likely an unexpected side effect of the deterioration in rotor speed tracking (which leads to less resonance between tower bending and the 3P rotor frequency) and should therefore be disregarded. An equal weighting realizes a good trade-off between all criteria. Presumably, this is because an equal weighting produces the most accurate wind speed estimate with the lowest variance and least high frequency content. Note, however, that under different tuning choices for the LIDAR low-pass filter and I&I estimator the optimal weighting may change.

**Interpretation of performance improvements.** We identify the following two main reasons for the superior performance of NOR+I&I+LIDAR. The first is NOR's unified design approach and seamless transition between regions 2 and 3, which is particularly enhanced by LIDAR. This is indicated by the significant improvements NOR+I&I+LIDAR achieves over ROSCO+LPFF near rated wind speed as shown in Figure 12, with ROSCO+LPFF even performing worse than ROSCO on tower fore-aft and blade flapwise DELs. We saw in Figure 13 that NOR+I&I+LIDAR achieves these improvements at region switching by a smoother transition that requires less extreme adjustments by the controls and consequently reduces fatigue loads. The second reason is the use of the averaged I&I and LIDAR estimate, where, as speculated in Section 3.3, this lower

| Lifetime | ROSCO | ROSCO+LPFF *cf.* ROSCO | NOR+I&I+LIDAR *cf.* ROSCO |
|---|---|---|---|
| Tower fore-aft DEL | 340.89 MNm | +0.35% | −1.53% |
| Tower side-to-side DEL | 324.96 MNm | −0.01% | +0.60% |
| Blade flapwise DEL | 51.75 MNm | +0.46% | −6.68% |
| Main shaft DEL | 7.78 MNm | −0.03% | −6.48% |
| Pitch rate | 0.035 °/s | +8.94% | −36.37% |
| Average power | 7.683 MW | −0.02% | +0.30% |
| Rotor speed RMS error | 0.084 rpm | −9.03% | −8.87% |
| Maximum rotor speed | 9.206 rpm | −3.61% | −5.66% |

**Table 6.** Lifetime average performance metrics of (feedback-only) ROSCO, and percentage change of ROSCO+LPFF and NOR with averaged I&I and LIDAR compared to ROSCO. With the exception of average power, a reduction is better.

| At $18\,\mathrm{m\,s^{-1}}$ | ROSCO | ROSCO+LPFF *cf.* ROSCO (Fu et al. (2023)) | NOR+I&I+LIDAR *cf.* ROSCO |
|---|---|---|---|
| Tower fore-aft DEL | 303.02 MNm | −6.99% (−4.3%) | −12.38% |
| Tower side-to-side DEL | 176.14 MNm | +6.58% | +1.18% |
| Blade flapwise DEL | 71.51 MNm | −3.19% (−2.9%) | −6.92% |
| Main shaft torque DEL | 1.87 MNm | −49.40% | −49.85% |
| Pitch rate | 0.069 °/s | +26.63% | −35.24% |
| Average power | 14.97 MW | +0.04% | −0.27% |
| Rotor speed RMS error | 0.441 rpm | −41.94% (−44.6%) | −37.54% |
| Maximum rotor speed | 9.206 rpm | −3.61% | −5.66% |

**Table 7.** Performance metrics of feedback-only ROSCO, and percentage change of ROSCO+LPFF and NOR+I&I+LIDAR compared to ROSCO, at mean wind speed 18 m/s. The numbers reported in Fu et al. (2023) are given in parantheses.

variation estimate indeed significantly reduces actuator usage and fatigue loads. The use of this average for NOR versus the traditional LIDAR signal for ROSCO+LPFF is the main difference between these controllers (see Section 4.4). This means that the significant improvements NOR+I&I+LIDAR achieves over ROSCO+LPFF at high wind speeds (as was shown in Table 7) are most likely due to this averaging.

Some further comparisons of the ROSCO and the NOR controllers from a systems theory perspective are included in the Appendices.

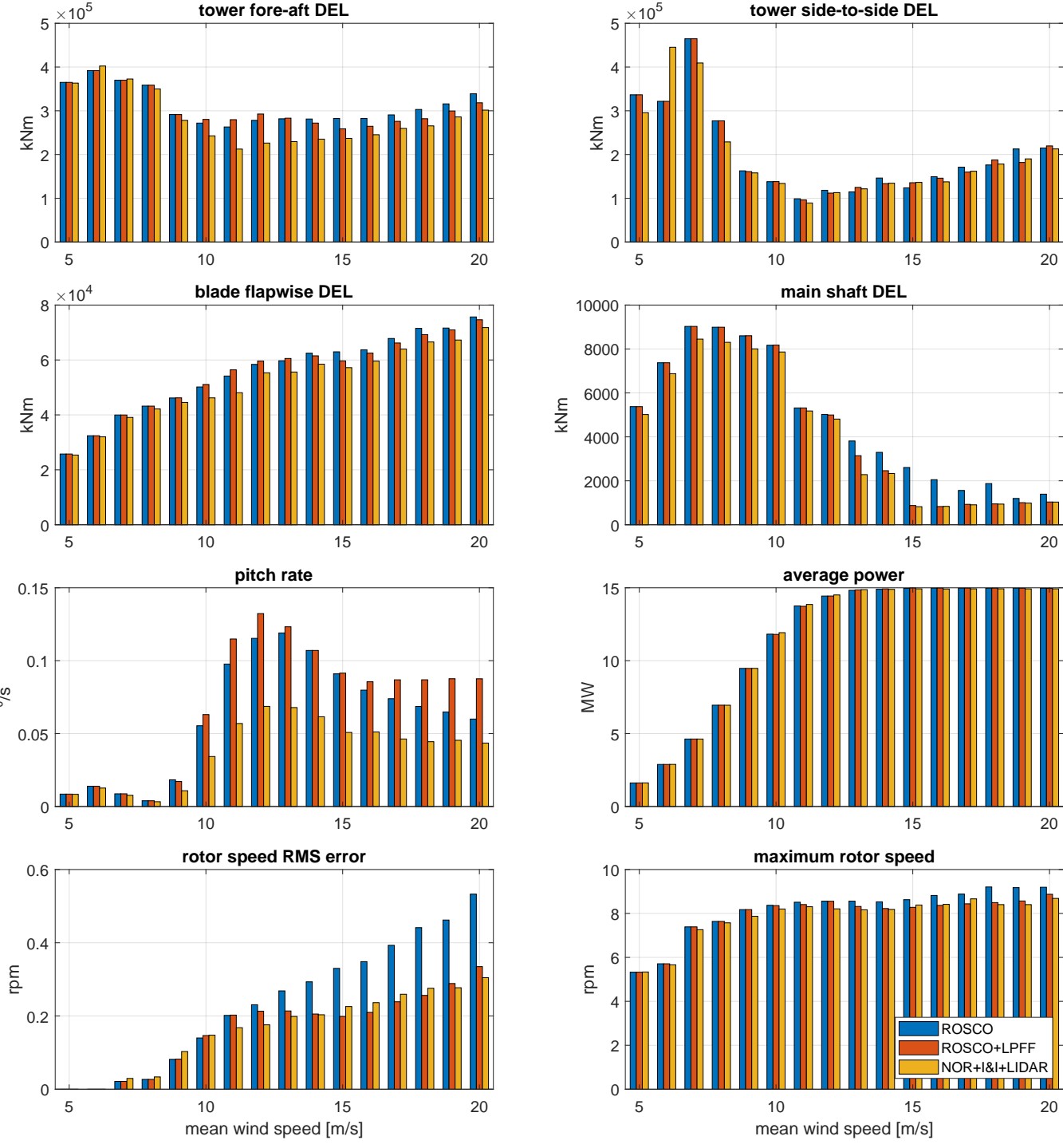

**Figure 12.** Performance comparisons between feedback-only ROSCO (blue), ROSCO with LIDAR pitch feedforward (red) and NOR+I&I+LIDAR with equal weighting from 2 hour simulations at each wind speed on the IEA 15-MW turbine.

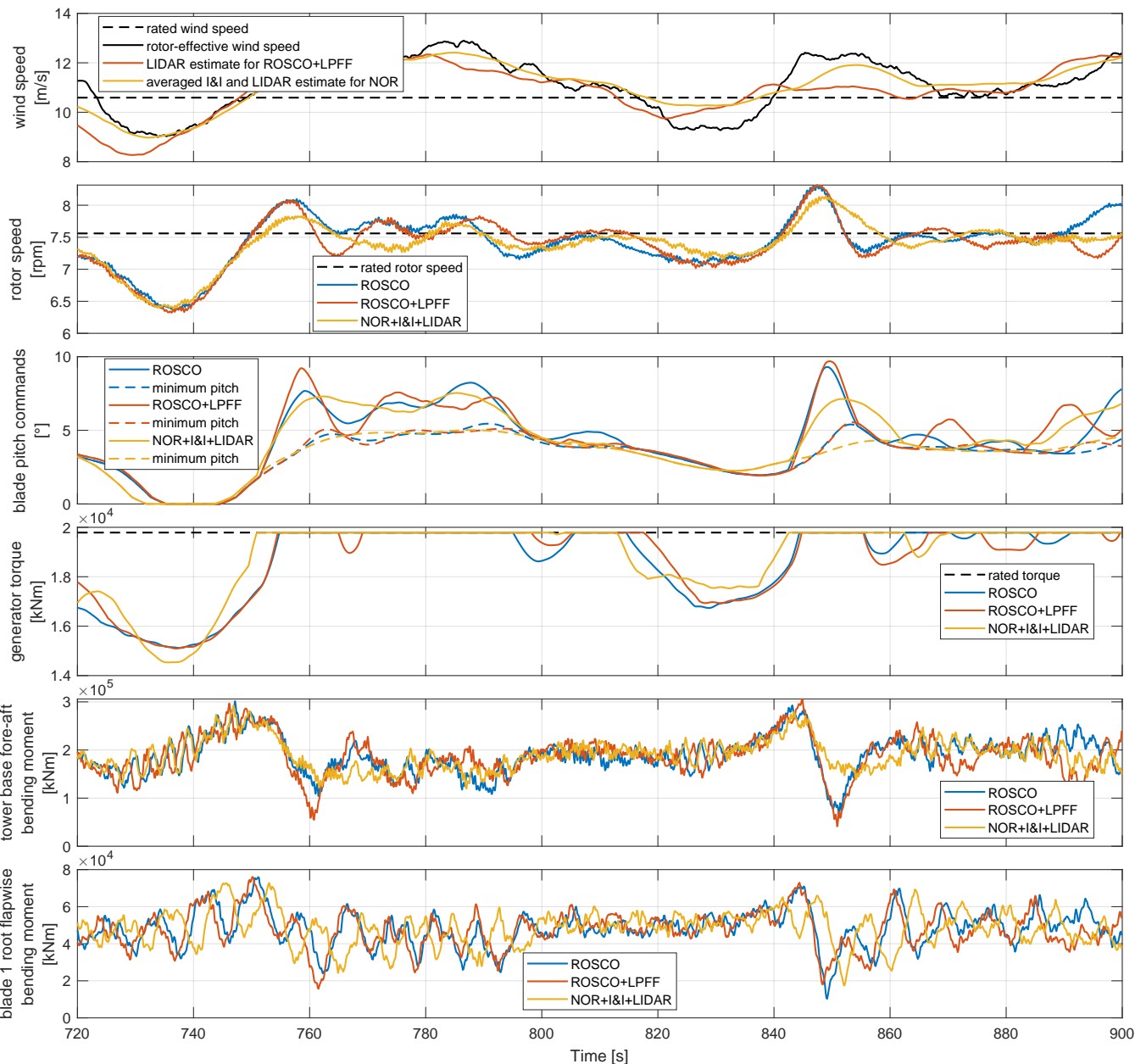

**Figure 13.** Comparison between feedback-only ROSCO (blue), ROSCO with LIDAR pitch feedforward (red), and NOR using the average between I&I and LIDAR estimates (yellow). Shown is a 3 minute interval of the simulation at mean wind speed $12\text{m/s}$. The minimum pitch commands $\theta_{\min}$ are due to peak shaving. Notice how at 740 and 840s NOR+I&I+LIDAR uses the LIDAR preview information in Region 2 to increase generator torque sooner than ROSCO and ROSCO+LPFF. This reduces subsequent blade pitch peaks in Region 3 at 760 and 850s, which in turn reduces the drop in tower base for-aft and blade root flapwise bending moments, and ultimately reduces DELs. Also note how blade pitch and generator torque controls are overall less oscillatory for NOR+I&I+LIDAR compared to ROSCO and ROSCO+LPFF. The different region switching behaviour can be observed as well: Where ROSCO's set point smoothing creates short transitional windows in which both or neither the minimum blade pitch and maximum generator torque saturations are active, NOR always activates exactly one.

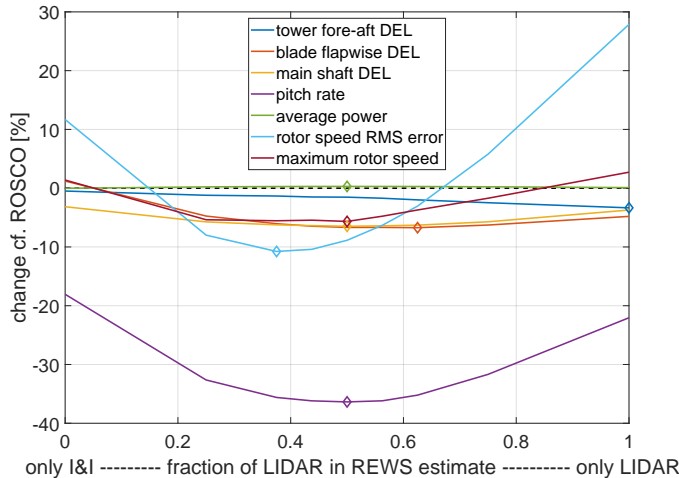

**Figure 14.** Comparison of different weightings of I&I and LIDAR for the NOR controller (see Section 3.3), with $\alpha = 0$ (left) to $\alpha = 1$ (right). The optima are marked for each performance criterion.

## 7 Conclusions

We have introduced a novel nonlinear controller design methodology for wind turbine control that utilises wind speed estimation that may be derived from the turbine's SCADA, LIDAR measurements, or both. NOR is based on the simple idea that
the turbine, modelled as a first order differential equation, should follow desired stable dynamics. In Region 2 the blade pitch angles are fixed to maximize power subject to peak shaving, while in Region 3 the generator torque is fixed to rated. The remaining one of these two is then chosen to solve a torque balance equation that realizes the desired dynamics, assuming that the wind speed estimate is perfectly accurate. Region switching is designed such that the torque balance equation always has a unique solution, the control signals do not jump when switching and always exactly one of them is saturated. We further
introduced the averaging of I&I and LIDAR estimates to create a higher quality low-variation wind speed estimate. Extensive simulation studies over a broad range of mean wind speeds and performance metrics showed that the NOR+I&I+LIDAR controller matches ROSCO+LPFF's capabilities of improving rotor speed tracking, but also significantly reduces fatigue loads and actuator usage.

From a design perspective, NOR has several advantages over ROSCO. It utilises a simple nonlinear turbine model, and hence
avoids the need for gain-scheduling methods based on a range of linearisation points. It provides a unified design approach over both operating regions. The closed loop maintains the desired dynamics, and Region 2 torque and Region 3 pitch controller transition in a continuous fashion. In this respect, NOR is also an improvement over the earlier gain-scheduled linear output regulation methods used in (Mahdizadeh et al., 2021) and (Woolcock et al., 2023). NOR enables direct and seamless inclusion of LIDAR wind speed estimates across operating regions, which ROSCO cannot do as easily. This, combined with the high
quality of the averaged I&I and LIDAR estimate leads to the significant performance improvements of NOR+I&I+LIDAR.

While in this work the NOR controller was designed to achieve maximum power point tracking, the controller can be adapted to output a different desired power. Thus future work can consider an application of NOR to problems of active power generation to provide grid frequency support services.

## Appendix A: Correction of modelling errors in NOR+I&I

The combination of NOR and I&I is robust towards errors in the modelling of the aerodynamic torque, in the sense that there are no asymptotic tracking errors even if there is a model mismatch. The reasoning for this is presented in the following.

By substituting equation (34) or (36) into (29), it follows that

$$J\dot{\Omega} = M_a(\Omega, v_x, \theta) - M_a(\Omega, \hat{v}_x, \theta) + J\mu(\Omega_{\text{ref}} - \Omega). \tag{A1}$$

Generally, the NOR controller, because it relies on an aerodynamic torque model $M_a(\Omega, v_x, \theta)$, would be susceptible to dis-
590 crepancies of that model from reality. However, the combination with the I&I estimator, which uses the same model for its wind speed estimate, compensates such errors. To see this, let $M_a(\Omega, v_x, \theta)$ be the real aerodynamic torque that the wind applies, and $\hat{M}_a(\Omega, \hat{v}_x, \theta)$ be the estimated aerodynamic torque that NOR and I&I use, with $\hat{M}_a$ indicating a different model from $M_a$, mainly due to biases in the Cp surface. The dynamic equation (22) for the I&I estimator then becomes

$$\dot{\hat{v}}_x = \frac{\kappa}{J}(M_a(\Omega, v_x, \theta) - \hat{M}_a(\Omega, \hat{v}_x, \theta)). \tag{A2}$$

Assume now that the wind speed $v_x$ is constant. Then $\hat{v}_x$ converges to some value $\hat{v}_x^\infty$, which is generally different from $v_x$. From (A2) it follows that

$$M_a(\Omega, v_x, \theta) = \hat{M}_a(\Omega, \hat{v}_x^\infty, \theta). \tag{A3}$$

Under the different aerodynamic torque model $\hat{M}_a$ the closed loop (A1) becomes

$$J\dot{\Omega} = M_a(\Omega, v_x, \theta) - \hat{M}_a(\Omega, \hat{v}_x, \theta) + J\mu(\Omega_{\text{ref}} - \Omega). \tag{A4}$$

Using (A3) it follows that in the limit the closed loop has the dynamics

$$J\dot{\Omega} = J\mu(\Omega_{\text{ref}} - \Omega), \tag{A5}$$

which are stable and achieve perfect reference tracking despite the model discrepancy.

Intuitively, this robustness can be explained in the following way: If, for example, the power coefficients in $\hat{M}_a$ are higher than in reality, the I&I estimator underestimates the wind speed. The NOR controller computes the generator torque based on
underestimated wind speed and overestimated power coefficient. These two effects compensate, and the correct aerodynamic torque is compensated.

## Appendix B: Theoretical comparison between NOR and ROSCO

The combination of NOR controller and I&I estimator is a form of "pseudo-feedforward". This is because the I&I estimate is the output of the estimator dynamic system, which has the rotor speed as input. Thus a feedforward of the I&I estimate works similarly to a PI controller. This means that NOR+I&I is a essentially a feedback controller like ROSCO, formulated as feedforward.

In this appendix a connection between NOR+I&I and ROSCO is established, which allows for comparison and even conversion of the control architectures. With linearization, both lead to second order closed-loop dynamics, and the parameters of the controllers can be converted into one another. Then, NOR+I&I and ROSCO are fundamentally the same controller. However, as pointed out before, NOR has advantages in the region switching and inclusion of LIDAR. A similar linearization analysis can be done to compare NOR+I&I+LIDAR with ROSCO+LPFF.

### B1 Linearized NOR in either region

Consider an equilibrium $\Omega^*, v_x^*, \theta^*, M_g^*$ of (6). Denote the partial derivatives of the aerodynamic torque by

$$\alpha = \left.\frac{\partial M_a(\Omega^*, v_x, \theta^*)}{\partial v_x}\right|_{v_x^*}, \tag{B1}$$

$$\beta = \left.\frac{\partial M_a(\Omega^*, v_x^*, \theta)}{\partial \theta}\right|_{\theta^*}, \tag{B2}$$

$$\gamma = \left.\frac{\partial M_a(\Omega, v_x^*, \theta^*)}{\partial \Omega}\right|_{\Omega^*}, \tag{B3}$$

and the deviations from the equilibrium by $\Delta\Omega = \Omega - \Omega^*$, etc. The linearizations of the plant model (6), the NOR control law (34) and (36) and the I&I estimator (22) are

$$J\dot{\Omega} = \alpha\Delta v_x + \beta\Delta\theta + \gamma\Delta\Omega - \Delta M_g, \tag{B4}$$

$$\Delta M_g = \alpha\Delta\hat{v}_x + \beta\Delta\theta + \gamma\Delta\Omega + J\mu\Delta\Omega - J\mu\Delta\Omega_{\text{ref}}, \tag{B5}$$

$$\dot{\hat{v}}_x = \tilde{\kappa}(\Delta v_x - \Delta\hat{v}_x). \tag{B6}$$

Note that this is the case no matter if the controller is operating in Region 2 or 3. Substituting (B5) into (B4) leads to

$$J\dot{\Omega} = \alpha(\Delta v_x - \Delta\hat{v}_x) + J\mu(\Delta\Omega_{\text{ref}} - \Delta\Omega) \tag{B7}$$

$$= \alpha\tilde{\kappa}^{-1}\dot{\hat{v}}_x + J\mu(\Delta\Omega_{\text{ref}} - \Delta\Omega). \tag{B8}$$

Differentiating (B7) and using (B8) to replace $\dot{\hat{v}}_x$ yields

$$J\ddot{\Omega} = \alpha\dot{v}_x - \alpha\dot{\hat{v}}_x + J\mu(\dot{\Omega}_{\text{ref}} - \dot{\Omega}), \tag{B9}$$

$$J\ddot{\Omega} = \alpha\dot{v}_x - \tilde{\kappa}J\dot{\Omega} + J\mu(\Delta\Omega_{\text{ref}} - \Delta\Omega) + J\mu\tilde{\kappa}(\dot{\Omega}_{\text{ref}} - \dot{\Omega}). \tag{B10}$$

Rearranging and dividing by $J$ gives the closed loop

$$\ddot{\Omega} + (\tilde{\kappa} + \mu)\dot{\Omega} + \tilde{\kappa}\mu\Delta\Omega = \alpha J^{-1}\dot{v}_x + \mu\dot{\Omega}_{\text{ref}} + \tilde{\kappa}\mu\Delta\Omega_{\text{ref}}. \tag{B11}$$

## B2   Linearized ROSCO in Region 2

In Region 2 ROSCO uses a gain-scheduled PI controller of the form

$$\Delta M_g = K_P^g (\Omega_{\text{ref}} - \Omega) + K_I^g \int (\Omega_{\text{ref}} - \Omega) \mathrm{d}t. \tag{B12}$$

Inserting this into the linearized plant equation (B4) and differentiation lead to

$$J\ddot{\Omega} = \alpha \dot{v}_x + \gamma \dot{\Omega} + K_P^g (\dot{\Omega} - \dot{\Omega}_{\text{ref}}) + K_I^g (\Omega - \Omega_{\text{ref}}), \tag{B13}$$

and rearranged,

$$\ddot{\Omega} - \frac{K_P^g + \gamma}{J}\dot{\Omega} - \frac{K_I^g}{J}\Delta\Omega = \frac{\alpha}{J}\dot{v}_x - \frac{K_P^g}{J}\dot{\Omega}_{\text{ref}} - \frac{K_I^g}{J}\Omega_{\text{ref}}. \tag{B14}$$

## B3   Linearized ROSCO in Region 3

Assume that the generator torque is set constantly to its rated value. The PI controller now takes the form

$$\Delta\beta = K_P^\theta (\Omega_{\text{ref}} - \Omega) + K_I^\theta \int (\Omega_{\text{ref}} - \Omega) \mathrm{d}t. \tag{B15}$$

Inserting this into (B4) and differentiation yield

$$J\ddot{\Omega} = \alpha \dot{v}_x + \gamma \dot{\Omega} + \beta K_P^\theta (\dot{\Omega}_{\text{ref}} - \dot{\Omega}) + \beta K_I^\theta (\Omega_{\text{ref}} - \Omega), \tag{B16}$$

and rearranged,

$$\ddot{\Omega} + \frac{\beta K_P^\theta - \gamma}{J}\dot{\Omega} + \frac{\beta K_I^\theta}{J}\Delta\Omega = \frac{\alpha}{J}\dot{v}_x + \frac{\beta K_P^\theta - \gamma}{J}\dot{\Omega}_{\text{ref}} + \frac{\beta K_I^\theta}{J}\Delta\Omega_{\text{ref}}. \tag{B17}$$

## B4   Parameter tuning for desired closed loop

We found that NOR+I&I and ROSCO lead to second order closed loop dynamics when linearized. This closed loop can be tuned using the controller and estimator gains. Let the closed-loop undamped natural frequency be $\omega_{\text{des}}$, and the damping factor be $\zeta_{\text{des}}$. The desired left-hand side is then

$$\ddot{\Omega} + 2\zeta_{\text{des}}\omega_{\text{des}}\dot{\Omega} + \omega_{\text{des}}^2 \Delta\Omega. \tag{B18}$$

The equations to tune the parameters of both controllers, obtained from equating this to the left-hand sides of (B11), (B14) and (B17), are then, for Region 2:

$$\begin{aligned} 2\zeta_{\text{des}}\omega_{\text{des}} &= \tilde{\kappa} + \mu = -\frac{K_P^g + \gamma}{J}, \\ \omega_{\text{des}}^2 &= \tilde{\kappa}\mu = -\frac{K_I^g}{J}. \end{aligned} \tag{B19}$$

And for Region 3:

$$\begin{aligned} 2\zeta_{\text{des}}\omega_{\text{des}} &= \tilde{\kappa} + \mu = \frac{\beta K_P^\theta - \gamma}{J}, \\ \omega_{\text{des}}^2 &= \tilde{\kappa}\mu = \frac{\beta K_I^\theta}{J}. \end{aligned} \tag{B20}$$

Note that the special case $\zeta_{\text{des}} = 1$ leads to $\tilde{\kappa} = \mu = \omega_{\text{des}}$.

With these equations, parametrizations of NOR+I&I and ROSCO can be converted into one another, such that the linearized closed loops are the same. The two controllers are then, if looked at in one Region in isolation, two sides of the same coin. While ROSCO approaches the nonlinearity with gain-scheduling, where the gains are obtained from differentials of the aerodynamic torque, NOR+I&I directly works with the nonlinear aerodynamic torque model. By chain rule, the controllers result in the same closed loop.

*Code and data availability.*  Code and data are available from http://doi.org/10.5281/zenodo.14523056

*Author contributions.*  RM developed the NOR controller, the idea of averaging I&I and LIDAR estimates and the theoretical foundations discussed in this article. He also contributed the majority of the code and conducted the simulations. RM was also the primary contributor to the writing of the article. RS served in a supervisory role, proposed the research problem discussed in this article, and contributed to the writing and revising of the article.

*Competing interests.*  The authors declare that they do not have any competing interests.

*Acknowledgements.*  The authors would like to thank Luke Woolcock for many fruitful discussions and guidance in developing the simulation framework. We would also like to thank the anonymous reviewers for their helpful comments that led to numerous improvements in the paper.

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
