# Peer review of "LIDAR-assisted nonlinear output regulation of wind turbines for fatigue load reduction"

_Wind Energy Science, 2024_

## Author Response (AR1)

**Authors' Response to the Reviewer Comments on Manuscript WES-2024-184**

May 5, 2025

We thank the reviewers for the time and effort they have spent on our manuscript and for their helpful suggestions. We changed numerous aspects of the manuscript, the most significant of which was that we now compare NOR not only to ROSCO, but also an existing LIDAR assisted control method that uses a region 3 blade pitch feedforward (referred to as ROSCO+LPFF in the manuscript). We also improved NOR+I&I+LIDAR based on reviewers' suggestions. We redid all the simulations and concluded that NOR+I&I+LIDAR significantly reduced fatigue loads and actuator use, compared to both ROSCO and ROSCO+LPFF. We further took great care in more clearly articulating the contributions of the paper across the manuscript. All numberings of lines, sections and figures in the responses below refer to the tracked changes version. We hope the revised version is now suitable for publication and look forward to hearing from you.

**Response to the Reviewer RC1**

**Major comments**

- **Comment 1** *Overall, the authors provide a good review of turbine control and an interesting concept that uses a combination of a wind speed estimate (WSE) and a LIDAR measurement to inversely solve for the ideal blade pitch and generator torque controls of the wind turbine.*

  **Response.** Thank you for the time and effort that you have spent on our manuscript.

- **Comment 2** *Many components of the controller (and the ROSCO baseline) are described in great detail but are not novel contributions in this article. Could those sections be streamlined for this audience?*

  **Response.** We shortened parts of Sections 2.5 and 4.2, while still including the introductions of concepts that are essential for controller design and keeping the manuscript accessible to a wider audience.

  **Changes in the manuscript.** lines 237-242, 511-516.

- **Comment 3** *The controller relies heavily on the $C_p$ surface. It's an interesting result that using the same $C_p$ surface in the wind speed estimator can relieve biases in the $C_p$ surface. Are there constraints (smoothness, monotonicity) that are required to get a unique pitch and torque at each time step?*

  **Response.** We thank the reviewer for raising this important question, which may not have been addressed sufficiently clearly in the first submission. The generator torque in the NOR controller is always given through an explicit formula, as well as the blade pitch angle in Region 2, so there are no issues with uniqueness and existence of solutions. For the case of finding blade pitch angles in Region 3, we have added Remark 4.1 to explain why the torque balance equation (36) always yields unique solutions. We also adjusted the abstract, introduction and Section 4.1 to better explain the NOR controller.

  **Changes in the manuscript.** lines 107-108, 407-415 (Remark 4.1), 427-434.

- **Comment 4** *It is interesting that the authors don't use any time delay or buffering in their LIDAR measurements. This can be a source of difficulty when using LIDAR measurements. I'm guessing that the controller performance then changes with wind speed because the lead time of the LIDAR changes, while the lag time of the WSE should be relatively constant. I wonder if you are reporting better performance at high wind speeds because the lead time of the LIDAR is lower and similar to the lag of the WSE.*

  **Response.** In the first submission, we noted in the concluding section that the LIDAR signal could potentially be improved by buffering with a time delay based upon the wind speed, but did not implement this. The reviewer's comment inspired us to investigate the improvements that might be obtained in this way, and these have been included in Section 3.3 of the revised paper as follows.

  Fu *et al* (2023) employed a buffer time given by equation (26) in the updated manuscript. For NOR+I&I+LIDAR, we used (27), decreasing the buffer time by the delay of the I&I estimator, in order to equalize the lead time of LIDAR and the lag time of I&I. We found that this more accurate REWS estimate improved performance compared to the version without buffering, and we updated the results section based on these new simulations. Furthermore, we added Figure 8, which illustrates how the averaged LIDAR and I&I estimate is obtained, including the buffered signal as in Fu *et al* (2023) as well as our modification.

  Finally, we now also study different weightings of I&I and LIDAR than just a simple average. For that the buffer time is modified to (28). See Figure 14 for the results and the newly added "Weighting of I&I and LIDAR" paragraph in Section 6 for a discussion.

  **Changes in the manuscript.** lines 330-357, 658-665.

- **Comment 5** *The reporting of results in time series form could be improved. Right now, the reader must scroll between figures and pages to evaluate long time series with subtle differences. Is there a short, illustrative section of time that demonstrates the difference between your NOR controller and ROSCO. One that shows the impact of LIDAR measurements? Another way to demonstrate differences is by binning the results in time and plotting against wind speed.*

  **Response.** We thank the reviewer for this helpful comment and have added several new figures to better illustrate the differences between the controllers. Figure 8, which we mentioned in our response to Comment 4, compares the different wind speed estimates. Figure 13 compares ROSCO, the LAC of Fu *et al* (2023), and NOR+I&I+LIDAR on a 3 minute interval. This demonstrates how NOR+I&I+LIDAR achieves significant reductions in tower and blade DELs and pitch rate. Figure 12 shows the results binned in time and plotted against mean wind speed.

  **Changes in the manuscript.** Figures 8 and 13 added.

- **Comment 6** *How exactly are you limiting the maximum thrust and fatigue loads to the levels in Section 2.5? What part of the controller (Algorithm 1) contributes to fatigue load reduction? The reduced variation in generator torque? A control engineer in practice will want to adapt these loads, and see the downstream effect on power. Are you able to do this with your control scheme? This would be an interesting result of the control concept.*

  **Response.** Regarding the limiting of the maximum thrust, this is achieved by the same peak shaving procedure used in ROSCO, where a minimum pitch schedule is calculated that limits thrust force below a certain threshold. The pitch angle output of the NOR controller is always at least this minimum pitch schedule; it is equal when the controller operates in Region 2 and higher when in Region 3 (due to the monotonicity of the $C_p$ surface). Note that all controllers in our performance comparisons employ peak shaving with the same minimum pitch schedule, hence this is not the reason for our reported fatigue load reductions.

  The fatigue load reductions are indeed obtained largely by reducing the variation in generator torque and blade pitch controls. This is achieved by NOR's seamless switching between Regions 2

and 3, where at any time exactly one of the torque and pitch controls is saturated, and these control signals are continuous in time across the region switching. Assuming that the wind speed estimate is perfectly accurate, the closed loop has the same first order stable dynamics $\dot{\Omega} = \mu(\Omega_{\text{ref}} - \Omega)$ it is designed for in either region. This is further enhanced by using LIDAR in both regions. When switching from Region 2 to 3, the torque control increases sooner due to LIDAR, and the pitch control is smoother as a consequence. This is illustrated in Figure 13, which we added as part of this revision, where the effect of this smoother torque and pitch actuation on tower and blade loads is apparent. Specifically the averaging of I&I and LIDAR creates a low variation estimate with little high frequency content, which leads to smoother pitch and torque controls and, consequently, lower DELs. We added the "Interpretation of performance improvements" paragraph to section 6 to discuss these aspects. We also adjusted introduction, Section 4.1 (on NOR) and the conclusion.

Lastly, we agree that it is desirable to adapt fatigue loads directly in the design of the controller, and then investigate their impact on the power generation. NOR permits a trade-off between rotor speed tracking performance and actuator usage by adjusting controller gains, as well as permitting trade-offs between thrust-related DELs and power sacrifice by adjusting maximum thrust in peak shaving. We added this explanation in Section 4.1.

**Changes in the manuscript.** lines 127-128, 348-353, 422-425, 666-676, 697-699.

**Minor comments**

- **Comment 7** *Line 175: there is, in most cases, a power sacrifice near rated power when using peak shaving control.*

  **Response.** We updated the formulation in Section 2.4 to clarify that there is a power sacrifice, but due to the flatness of the $C_p$-surface (see Figure 1) at low $\theta$ it is relatively small.

- **Comment 8** *Section 2.5: I don't understand your units on the fatigue loads. Are these DELs? They should have the same units as the load, N-m. What are the nominal loads of the baseline control? How do these compare?*

  **Response.** These values were intended to be typical maximum blade tip and tower top displacements (in metres, not DELs), to give the reader an idea of the extent of the bending. To avoid confusion, we removed these values.

- **Comment 9** *Results: it may make more sense to only compare RMS rotor speed error above rated wind speeds.*

  **Response.** We agree and changed our formula for the RMS error in Section 5 to only account for times when the REWS is above rated.

**Comment 10** *L498: Should $\hat{M}_a$ have $\hat{v}_x$ in it?*

**Response.** Our intention here was to introduce the function $\hat{M}_a$ of rotor speed, wind speed and blade pitch angle, which differs from the actual aerodynamic torque model $M_a$. Nevertheless, we changed it to $\hat{v}_x$ and added a short sentence to explain the role of $\hat{M}_a$.

**Editorial comments**

- **Comment 11** *openFAST should be OpenFAST.*

  **Response.** fixed.

- **Comment 12** *Figs 4 and 5 could be combined; they look very similar to the ones in the ROSCO paper.*

  **Response.** done.

- **Comment 13** *The capitalization and font size in figures should match the text. Also there are labels like "windspeed" that should be "Wind speed (m/s)," for example. Fig 5 has a legend with a variable, and it's not defined in the caption.*

  **Response.** We adjusted the capitalization in several figures. Many of the figures with oversized font were generated with the two column version in mind to be the size of one column. We will make sure to appropriately format the figures for the final version. Figure 5 from the first submission has been combined with Figure 4.

- **Comment 14** *When substituting equations, it could help the reader to add a few descriptive words referencing the equations from earlier.*

  **Response.** We made such adjustments in several places in the manuscript, particularly in the appendices.

We again thank the reviewer for their very insightful comments that have lead to numerous valuable improvements to the manuscript. We have provided an acknowledgement of their contribution in the Acknowledgements section.

**Response to the Reviewer RC2 Comments**

- **Comment 15** *The paper presents a wind turbine control design using the nonlinear output regulation (NOR) method to reduce fatigue loads. The topic is interesting and relevant, but the motivation could be more clearly articulated. Given that the authors have published similar work in 2021, it would be helpful to clarify the specific motivation for employing the NOR method in this study.*

  **Response.** We thank the reviewer for their generally supportive comments. Regarding our earlier work in this area, we added Remark 4.2, where the proposed NOR controller is compared to the EOR controller from our previous work, and the advantages are pointed out. We also adapted the abstract, introduction and conclusion to make the paper's novel contributions more apparent.

  **Changes in the manuscript.** lines 6-12, 92-115, 435-442, 681-687.

- **Comment 16** *The proposed controller maximises power output by tracking the rotor speed set-point, which is computed based on a static $C_p$ surface and the optimal tip-speed ratio. This approach closely resembles the traditional $K\Omega^2$ law. However, unlike the $K\Omega^2$ squared method, the proposed approach relies on estimated wind speed, introducing additional sources of uncertainty. From an industrial perspective, this could be a potential drawback compared to the traditional method.*

  **Response.** We acknowledge and agree that heavily relying on a model for a controller is undesirable, and that controllers should either be designed without the need of an exact model or to compensate model errors. Note that the constant $k$ in the $k\Omega^2$ method is computed from several parameters like the optimal tip speed ratio, optimal power coefficient and air density, which are subject to uncertainty. Any difference between reality and the model parameters may cause the $k\Omega^2$ controller to track a slightly wrong tip speed ratio. As our controller heavily uses the $C_p$ surface, we understand the reviewer's concerns about the effect of model uncertainty.

  Additionally, when including LIDAR preview information, the LIDAR signal is subject to uncertainty and may have a bias compared to the actual REWS, i.e., usually be slightly higher or lower.

In fact, our initial simulations with NOR using LIDAR or the averaged I&I+LIDAR estimate led to significant errors, where Region 3 average power was around 16MW when it should be 15MW. This indicates that our $C_p$ surface (or LIDAR signal, or both) has significant bias.

Nonetheless, even with imperfect model information, NOR is able to achieve good control performance, for the following reasons. Firstly, the combination of NOR+I&I compensates errors in the $C_p$ surface, as detailed in Appendix A. Both I&I and NOR use the same $C_p$ model. Suppose the power coefficients are assumed too high, then the I&I estimator overestimates the aerodynamic torque caused by any given wind speed. This leads it to underestimate the wind speed based on the actual aerodynamic torque's effect on the rotor speed. The NOR controller now works with an underestimated wind speed estimate and an overestimated $C_p$ surface. These two compensate each other in the formula (4) for the aerodynamic torque, so that the torque and pitch controls (34) and (36) are computed based on the actual aerodynamic torque that the wind turbine experiences, even when the $C_p$ surface is biased.

Regarding biases in the LIDAR signal compared to REWS, we used the mean correction technique of the NOR+I&I+LIDAR controller, which is now described in the new Section 4.1.1. This extends the error compensating effect of NOR+I&I to NOR+I&I+LIDAR. The idea is that the resulting error corrected LIDAR signal has the same mean as the I&I estimate. This compensates any existing biases between LIDAR and the actual REWS, and makes it so that biases in the $C_p$ surface for the NOR controller are compensated. Our simulations with NOR+I&I+LIDAR show very accurate power tracking in Region 3 (see Figure 12 of the revised version), which confirms that accurate rotor speed tracking is achieved despite mismatches in the $C_p$ surface and the LIDAR signal.

We added several sentences in Section 4.1 (right above the newly added Remark 4.2) as well as Remark 4.3 in Section 4.1.1 to include these explanations in the manuscript.

**Changes in the manuscript.** lines 107-108, 426-434, 453-473.

- **Comment 17** *Additionally, the title suggests that the paper focuses on reducing fatigue loads, yet Algorithm 1 does not explicitly account for fatigue mitigation. Is this achieved through peak shaving or another mechanism? Providing further details on this aspect would strengthen the paper's contribution.*

**Response.** We apply the same form of peak shaving to all our controllers, therefore this is not the source of the fatigue load reduction that we found for NOR+I&I+LIDAR compared to ROSCO and the newly added LIDAR assisted controller (LAC). Fatigue mitigation is achieved by two things.

Primarily, the fatigue load reductions are obtained largely by reducing the variation in generator torque and blade pitch controls. This is achieved by NOR's seamless switching between Regions 2 and 3, where at any time exactly one of the torque and pitch controls is saturated, and these control signals are continuous in time across the region switching. Assuming that the wind speed estimate is perfectly accurate, the closed loop has the same first order stable dynamics $\dot{\Omega} = \mu(\Omega_{\text{ref}} - \Omega)$ it is designed for in either region.

This is further enhanced by using LIDAR in both regions. When switching from Region 2 to 3, the torque control increases sooner due to LIDAR; likewise in transitioning from Region 3 to 2, the pitch control decreases sooner. In both cases it is smoother as a consequence. This is illustrated in Figure 13, which we added as part of this revision, where the effect of this smoother torque and pitch actuation on tower and blade loads is apparent. Specifically the averaging of I&I and LIDAR creates a low variation estimate with little high frequency content, which leads to smoother pitch and torque controls and, consequently, lower DELs.

We added the "Interpretation of performance improvements" paragraph to section 6 to discuss these aspects. We also adjusted several other sections of the manuscript to make these facts clearer, particularly in the introduction, Section 4.1 (on NOR) and the conclusion.

**Changes in the manuscript.** lines 6-12, 422-425, 666-676, 681-687.

- **Comment 18** *The paper also states that averaging the estimated wind speed with LIDAR measurements improves the estimation of low-variation real-time wind speed. It would be helpful to elaborate on why simple averaging was chosen over a weighted sum. What was the motivation behind this decision?*

  **Response.** In the first submission, we used a simple average for simplicity, but as part of this revision we studied the effect of different weightings. This is now theoretically introduced at the end of Section 3.3. Figure 14 shows the effect of different weightings of I&I and LIDAR for NOR on performance. Overall, equal weighting performs best. This is likely due to the I&I and LIDAR estimates having similar variance/high frequency energy. When taking a weighted mean between two identically independently distributed random variables, the variance of the mean is minimized at equal weighting. Presumably this is the reason why our equally weighted average works best, because it leads to minimal variance/high frequency energy in the wind speed estimate and consequently pitch and torque commands, which not only reduces actuator usage, but also leads to smoother control with less extreme peak torques and thrusts, which reduces fatigue loads (this is also demonstrated in the newly added Figure 13). Note that different tuning of the LIDAR system and low-pass filter and the I&I estimator may change the optimal weighting. Hence, we are not saying that equal weighting is always best per se, it just happens to be best for the tuning we used.

  The newly added "Weighting of I&I and LIDAR" paragraph in Section 6 discusses these results.

  **Changes in the manuscript.** 353-357, 658-665.

- **Comment 19** *Finally, while the results show that the proposed method outperforms ROSCO, this is perhaps expected given that it incorporates a DAC approach and LIDAR. A more informative comparison might be against other LIDAR-assisted control methods to better assess the advantages of the proposed approach.*

  **Response.** We thank the reviewer for this very helpful suggestion. We undertook a search of the recent literature on LIDAR assisted control (LAC) and found that the method of Fu *et al* (2023) and references therein, which adds a LIDAR-based pitch feedforward, would be suitable for including in out performance comparisons. Thus we have added Section 4.3 to briefly describe this LAC modification of ROSCO, referred to as ROSCO+LPFF in the manuscript. We also improved our I&I+LIDAR estimate based on this existing research, particularly buffering our LIDAR estimate as described in Section 3.3 of the revision. Our simulations show that NOR+I&I+LIDAR matches the ROSCO+LPFF in rotor speed regulation, but reduces pitch rate by around 36% and blade flapwise lifetime DEL by 6.7%, which, with a Wöhler exponent of 10, roughly doubles the lifespan. In all other lifetime performance metrics NOR+I&I+LIDAR matches or is superior to ROSCO+LPFF. We also compare the performance at mean wind speed of 18m/s in isolation; this is shown in Table 7 of the revised manuscript and discussed in the revised results section. We closely replicated the blade flapwise DEL and rotor speed tracking improvements that Fu *et al* (2023) reported for ROSCO+LPFF compared to feedback-only ROSCO. NOR+I&I+LIDAR roughly doubles these reductions in tower fore-aft and blade flapwise DEL, and also reduces pitch rate significantly. Hence, our proposed controller leads to significantly better performance than the existing LAC method in lifetime metrics as well as in Region 3 in isolation. The improvements are due to NOR's ability to use LIDAR preview information in both wind regions and in a smooth manner at the region switching. Furthermore, the I&I+LIDAR average is mainly responsible for the large reduction in pitch travel and part of the fatigue load reductions. We added the "Interpretation of performance improvements" paragraph in Section 6 to discuss this.

  **Changes in the manuscript.** A large portion of the changes highlighted in the tracked changes manuscript refers to the inclusion of ROSCO+LPFF.

- **Comment 20** *Given these points, I believe the paper would benefit from further clarification and refinement. I encourage the authors to address these concerns, as doing so would strengthen the manuscript significantly.*

**Response.** We thank the reviewer for their valuable suggestions. They have helped us to better expressing the contributions of our work, and we believe that adopting the suggestion to include a LAC method in our comparisons has provided further evidence of the performance improvements that the NOR method can provide, relative to control methodologies from the recent literature.

We again thank the reviewer for their very insightful comments that have lead to numerous valuable improvements to the manuscript. We have provided an acknowledgement of their contribution in the Acknowledgements section.